# Mef2C restrains microglial inflammatory response and is lost in brain ageing in an IFN-I-dependent manner

Aleksandra Deczkowska [1], Orit Matcovitch-Natan[1,2], Afroditi Tsitsou-Kampeli[1], Sefi Ben-Hamo[1], Raz Dvir-Szternfeld[1], Amit Spinrad[1,2], Oded Singer [3], Eyal David[2], Deborah R. Winter [2], Lucas K. Smith[4], Alexander Kertser[1], Kuti Baruch [1], Neta Rosenzweig[1], Anna Terem[5,6], Marco Prinz[7,8], Saul Villeda[4], Ami Citri [5,6], Ido Amit[2] & Michal Schwartz[1]

During ageing, microglia acquire a phenotype that may negatively affect brain function. Here we show that ageing microglial phenotype is largely imposed by interferon type I (IFN-I) chronically present in aged brain milieu. Overexpression of IFN-β in the CNS of adult wild-type mice, but not of mice lacking IFN-I receptor on their microglia, induces an ageing-like transcriptional microglial signature, and impairs cognitive performance. Furthermore, we demonstrate that age-related IFN-I milieu downregulates microglial myocyte-specific enhancer factor 2C (Mef2C). Immune challenge in mice lacking Mef2C in microglia results in an exaggerated microglial response and has an adverse effect on mice behaviour. Overall, our data indicate that the chronic presence of IFN-I in the brain microenvironment, which negatively affects cognitive function, is mediated via modulation of microglial activity. These findings may shed new light on other neurological conditions characterized by elevated IFN-I signalling in the brain.

[1] Department of Neurobiology, Weizmann Institute of Science, Rehovot 7610001, Israel. [2] Department of Immunology, Weizmann Institute of Science, Rehovot 7610001, Israel. [3] Faculty of Biochemistry, Life Sciences Core Facilities, Weizmann Institute of Science, Rehovot 7610001, Israel. [4] Department of Anatomy, University of California San Francisco, San Francisco, CA 94143, USA. [5] Department of Biological Chemistry, Institute of Life Sciences, Faculty of Natural Sciences, The Hebrew University, Jerusalem 91904, Israel. [6] Edmond and Lily Safra Center for Brain Sciences, The Hebrew University, Jerusalem 91904, Israel. [7] Institute of Neuropathology, Faculty of Medicine, University of Freiburg, Freiburg 79106, Germany. [8] BIOSS Centre for Biological Signalling Studies, University of Freiburg, Freiburg 79104, Germany. Aleksandra Deczkowska and Orit Matcovitch-Natan contributed equally to this work. Ido Amit and Michal Schwartz jointly supervised this work. Correspondence and requests for materials should be addressed to I.A. (email: Ido.Amit@weizmann.ac.il) or to M.S. (email: Michal.Schwartz@weizmann.ac.il).

The activity of microglia, the highly specialized resident myeloid cells of the central nervous system (CNS), is essential for brain and spinal cord development, maintenance, protection and repair throughout life. Studies suggest that the microglial phenotype is distinct from that of other tissue-resident macrophages and is highly dependent on the unique, heterogeneous and dynamic microenvironment of the CNS[1–3].

Throughout life, microglia routinely perform a range of homoeostatic activities critical for CNS maintenance: they shape neuronal circuitry and synaptic function, mediate removal of debris and apoptotic bodies, produce trophic factors that support neuronal activity, and serve as the immune sentinels of the CNS[4–7]. Since immune activities in the brain could be detrimental to the vulnerable neuronal tissue, microglial responses to immune stimuli are maintained under strict regulation[8]. Such immune-restraints, or 'off' signals, are mediated by direct interactions between microglia and neurons through the receptor-ligand pairs CX$_3$CL1-CX$_3$CR1 and

CD200-CD200R, by the soluble milieu of the CNS, especially TGFβ, as well as by internal signalling regulators, such as the transcription factor MafB[1, 9–11]. Nevertheless, during ageing, microglia not only lose their supportive phenotype, but also acquire activities that might exacerbate ageing-related brain function decline; the understanding of why and how these changes in microglial phenotype occur is in its infancy.

Here we investigate the aged microglial phenotype, and find that it is largely determined by interferon type I (IFN-I), a factor chronically expressed by the choroid plexus during ageing[12]. Persistent induction of IFN-I signalling in adult mouse brain causes decreased cognitive performance. This effect is mediated by microglia, which in response to chronic IFN-I assume an ageing-like transcriptional phenotype. Furthermore, we find that Mef2C is negatively regulated by IFN-I, and functions as a microglial 'off' factor, conferring resilience to inflammatory conditions, which prevail in the aged brain.

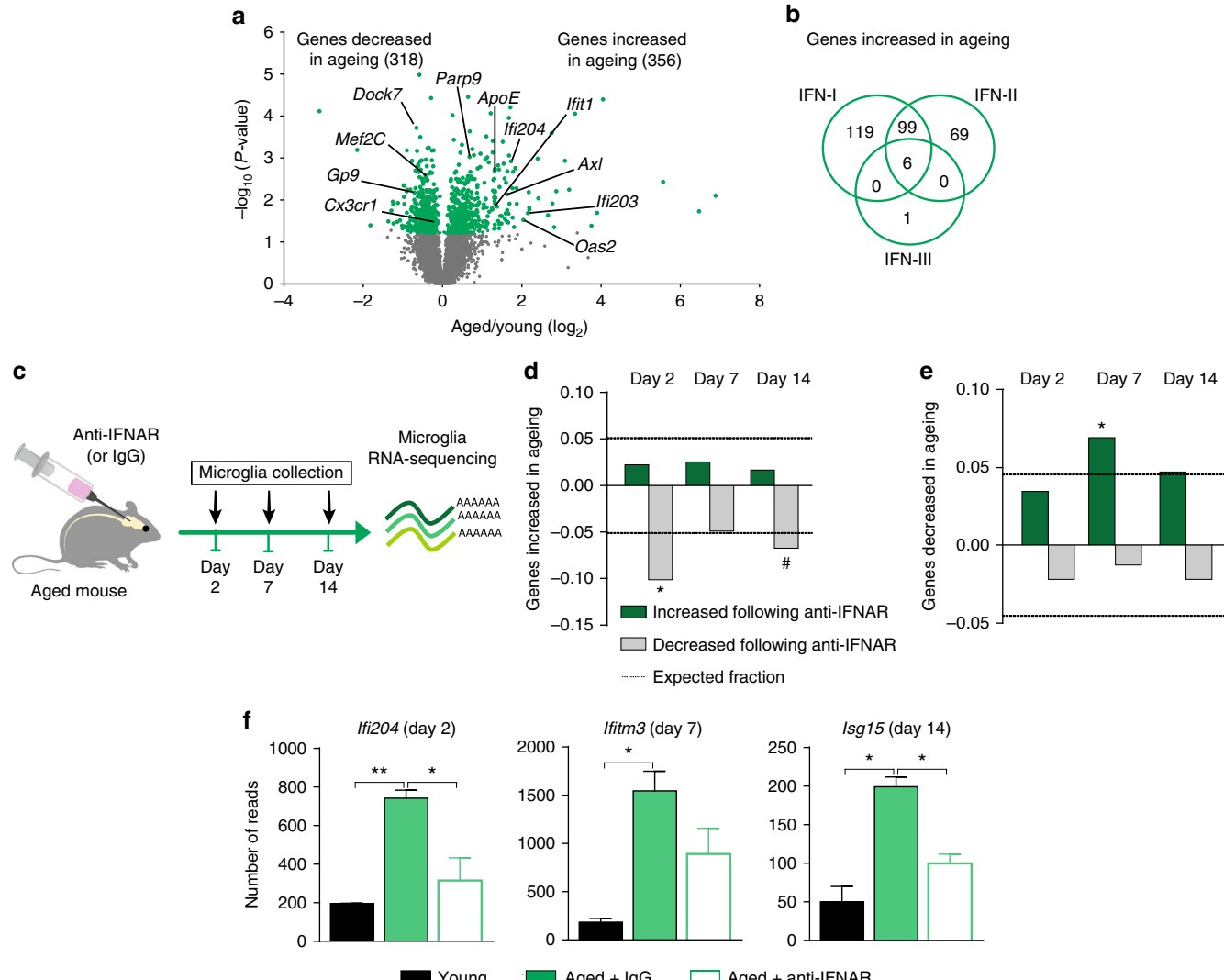

**Fig. 1** IFN-I in ageing brain microenvironment induces ageing microglial phenotype. **a** Volcano plot showing the fold change and significance of genes between microglia of young (3 months old) and aged (22 months old) mice ($n = 3$, $P < 0.05$; unpaired t test). **b** Venn diagram illustrating dependence of genes significantly upregulated in aged microglia on IFN-I, -II and -III, based on the Interferome 2.01 database. **c** Schematic presentation of the IFN-I receptor (IFNAR) blockade experiment in aged mice. **d**, **e** Fraction of differentially expressed genes at each time-point following IFNAR blockade, that showed either an increase (**d**) (*$P = 3.2E$-12, #$P = 2.6E$−05) or a decrease (**e**) (*$P = 0.0001$, hypergeometric distribution) in aged microglia (relative to young). *Dashed line* indicates the expected distribution of genes. **f** mRNA expression levels of *Ifi204*, *ifitm3* and *Isg15* in microglia of young and aged anti-IFNAR or IgG-treated mice. **$P < 0.01$, *$P < 0.05$; one-way ANOVA with Newmann–Kleus post hoc test, $n = 3$ per group, the results are representative of two independent experiments

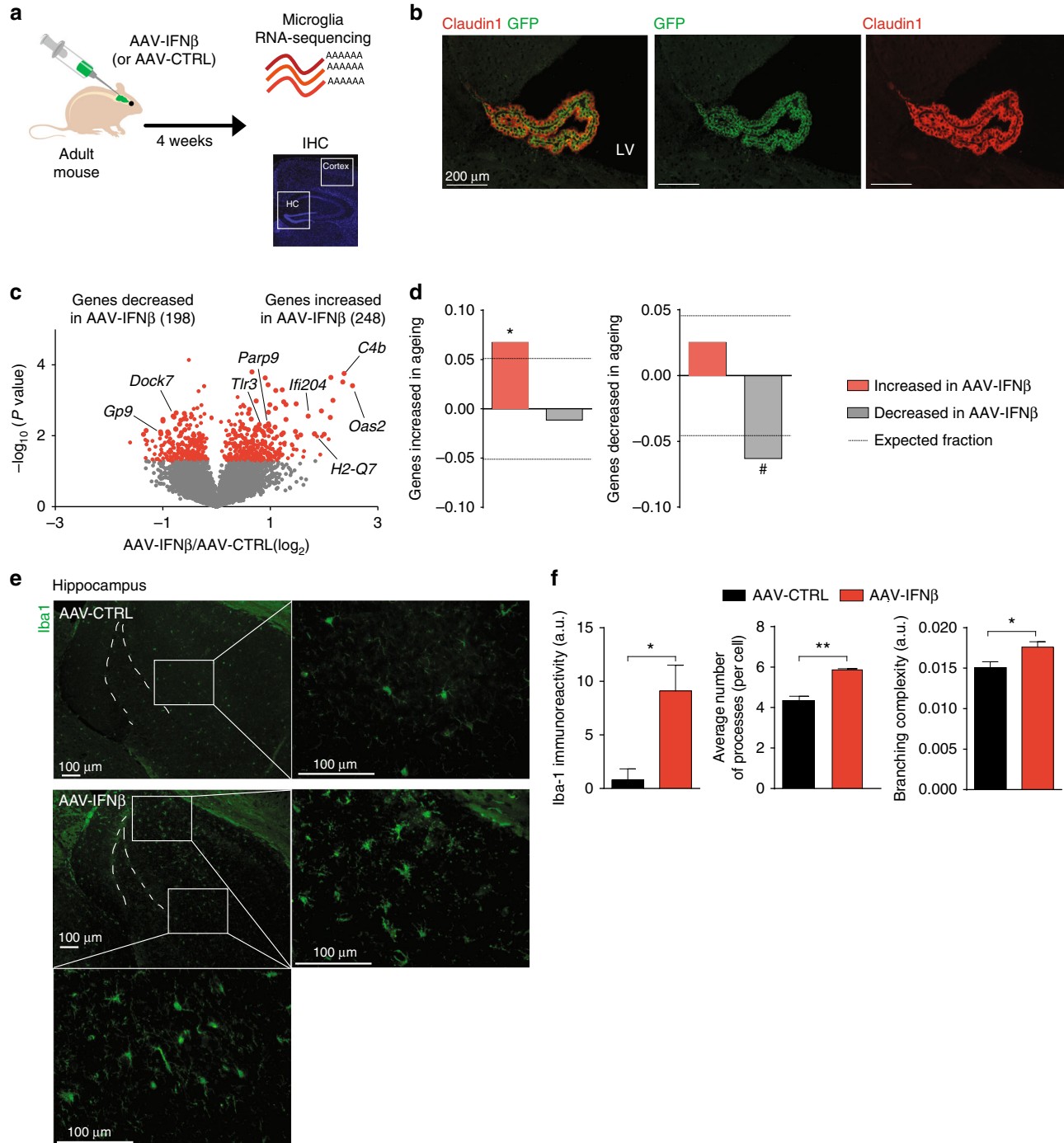

**Fig. 2** Microglia of adult mice express an ageing-like phenotype following chronic exposure to IFN-β. **a** Schematic presentation of the experimental design for testing the effect of IFN-I overexpression on microglia. **b** Representative image of the choroid plexus in the lateral ventricle (LV; contralateral to the intracerebroventricular administration site), expressing GFP following AAV-mediated infection (GFP in *green*, Claudin-1 marking epithelial tight junctions in *red*, nuclear Hoechst staining in *blue*). **c** Volcano plot showing the fold change and significance of gene expression values between microglia from IFN-I-overexpressing (AAV-IFNβ) and control (AAV-CTRL) adult mice ($P < 0.05$; unpaired $t$-test, $n = 2$–3 per group). **d** Fraction of differentially expressed genes from the microglia of IFN-I-overexpressing mice that were increased (*left*) or decreased (*right*) in aged mice (relative to young). *Dashed line* indicates the expected distribution of genes. Significance of enrichment or depletion *$P = 0.0008$, #$P = 0.0011$; hypergeometric test. **e** Representative micrographs of Iba-1 staining in microglia in the hippocampus of mice infected with AAV-IFNβ or AAV-CTRL. **f** Quantification of Iba-1 staining intensity and analysis of microglial morphology reveals IFN-I-induced microgliosis in the hippocampus. **$P < 0.01$, *$P < 0.05$; unpaired $t$-test, $n = 5$ per group, the results are representative of two independent experiments

## Results

**The microglial signature in ageing is imposed by IFN-I.** Our first goal was to gain a non-biased insight as to how microglial cells are altered during ageing. To this end, we sorted CD11b⁺CD45^int microglia (Supplementary Fig. 1a, b) from whole brains

of either young (3 months old) or aged (22 months old) wild-type C57Bl/6 J mice, and compared their transcriptional profile using RNA sequencing (RNA-Seq) (Fig. 1a, Supplementary Dataset 1). We identified 674 genes that were differentially expressed in microglia between young and aged mice. Gene Ontology (GO)

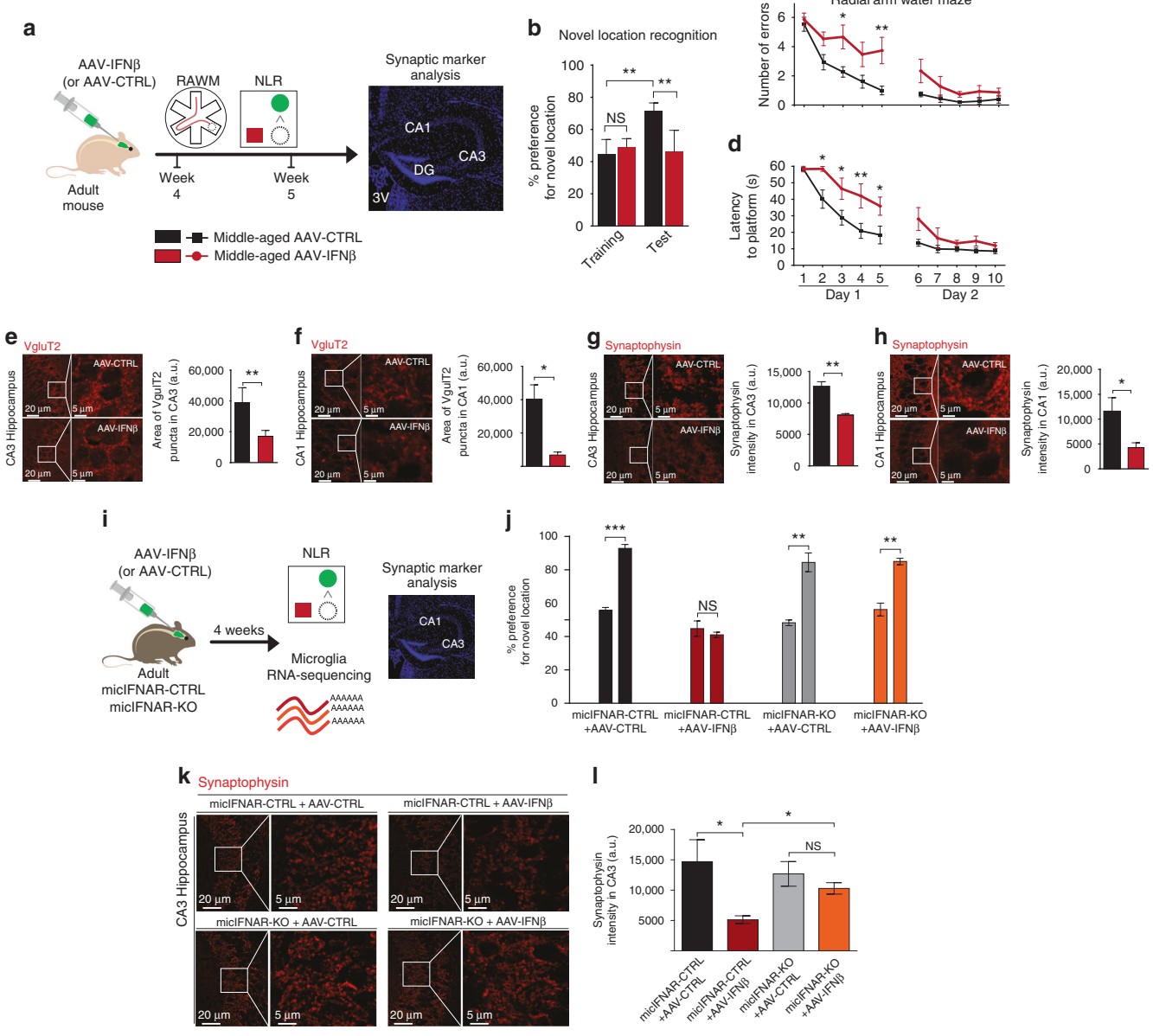

**Fig. 3** IFN-β in the brain negatively affects learning abilities in microglia-dependent manner. **a** Schematic presentation of the experimental design for testing the effect of IFN-I on cognitive ability **b** Performance of middle-aged AAV-infected mice in Novel Location Recognition task for assessment of hippocampal memory (one-way ANOVA with Newman–Keuls post hoc test; $n = 5$ per group). **c** Number of errors, and **d** latency to hidden platform in RAWM (two-way repeated-measures ANOVA with Bonferroni post-hoc test; $n = 5$ per group). **e**, **f** Representative micrographs and quantification of the presynaptic marker, VgluT2$^+$ puncta (*red*) in CA3 (**e**) and CA1 (**f**) of AAV-infected mice. **g**, **h** Representative micrographs and quantification of staining intensity of a presynaptic marker, Synaptophysin (*red*) in CA3 (**g**) and CA1 (**h**) of AAV-infected mice (unpaired *t*-test, $n = 5$ per group). **i** Schematic presentation of the experimental design for testing the involvement of microglia in IFN-I-dependent cognitive loss. **j** Performance of middle-aged AAV-infected mice mic-IFNAR-CTRL and mic-IFNAR-KO in NLR (paired *t*-test; $n = 4$–5 per group). **k**, **l** Representative micrographs (**k**) and quantification (**l**) of staining intensity of a presynaptic marker, Synaptophysin (*red*) in CA3 of AAV-infected mic-IFNAR-CTRL and mic-IFNAR-KO mice (one-way ANOVA with Newmann–Kleus post hoc test; $n = 4$ per group). In all panels *$P < 0.05$; **$P < 0.01$, the results are representative of two independent experiments

enrichment analysis of genes upregulated in microglia of aged animals indicated "immune system process" and specifically, "response to virus" among the most significantly enriched terms (Supplementary Fig. 2a–c). Since many of these genes were annotated to interferon signalling, related to anti-viral responses, we systematically searched for IFN pathways using Interferome 2.01 (http://www.interferome.org/interferome/search/showSearch.jspx) and found that 63% (224 out of 356) of the genes significantly upregulated in ageing are part of the IFN-I signalling response (Fig. 1b).

It was previously shown that in ageing, IFN-β is chronically produced by the choroid plexus, a monolayer of epithelial cells at the blood-cerebrospinal fluid barrier[12, 13]. We therefore hypothesized that ageing microglial phenotype might be imposed by the IFN-I released from the aged choroid plexus to the cerebrospinal fluid (CSF). To test this hypothesis, we administered neutralizing antibodies directed against the IFN-I receptor (anti-IFNAR) or isotype control antibodies (IgG) to the CSF of aged mice. Whole-brain microglia were sorted 2, 7 or 14 days following the antibody administration, and their gene expression profile was analysed

(Fig. 1c). We found that IFNAR blockade resulted in changes in global gene expression that were inversely related to those found in the ageing phenotype at all three time-points (Fig. 1d–f, Supplementary Fig. 3a–c, Supplementary Dataset 2).

**Chronic IFN-β induces an ageing microglial phenotype.** To isolate the effect of IFN-β on microglia, relative to other ageing-associated factors, we established an in vivo system for long-term overexpression of IFN-β at the choroid plexus in young mice. To this end, we created an adeno-associated virus (AAV1) carrying murine *Ifnb* and *eGFP* (AAV-IFNβ), while the same vector without the *Ifnb* insert was used as a control (AAV-CTRL). In vitro analysis revealed that infection with AAV-IFNβ, but not with AAV-CTRL, led to the production of biologically active murine IFN-β cytokine (Supplementary Fig. 4a). To validate the spatial specificity of this approach in vivo, we infected the choroid plexi of young mice by intracerebroventricular injection of AAV-IFN-β or AAV-CTRL (Fig. 2a). Immunostaining and RT–qPCR analysis of *GFP* expression in the AAV-infected brains indicated that both viruses infected the choroid plexus but not the brain parenchyma (Fig. 2b, Supplementary Fig. 4b, c), in accordance with previous reports[14, 15]. Further, RT-qPCR analyses of IFN-I-dependent genes in the CP, hippocampus and spleen of AAV-infected mice, confirmed that response to IFN-β in this model was confined to the choroid plexus (Supplementary Fig. 4d–f) and remained present at this site up to 7 months after infection (Supplementary Fig. 4g, h).

To determine whether the mere continuous presence of IFN-β in the CNS in young animals could be sufficient to induce an "ageing" microglial phenotype, we isolated whole-brain microglia from mice overexpressing IFN-β and controls, and analysed their transcriptional signature using RNA-Seq (Fig. 2c, Supplementary Dataset 3). Globally, we found a significant overlap between genes with increased expression in microglia isolated from AAV-IFNβ-infected mice and those elevated in microglia from aged mice, and between the genes with decreased expression in microglia of AAV-IFNβ and in aged animals (Fig. 2d). Moreover, the fold induction of IFN-I-dependent gene expression in microglia in AAV-IFNβ mice (relative to AAV-CTRL) and in ageing (relative to young animals) were comparable (Supplementary Fig. 4i, Supplementary Datasets 1 and 3), indicating that the AAV-mediated IFNβ overexpression model is suitable to further study the interactions between IFN-I signalling, microglia and brain function.

Analysis of microgliosis in the infected mice revealed a spatially diverse effect of IFN-I. Hippocampal microglia, located physically closer to the source of IFN-I in AAV-IFNβ mice showed markedly increased Iba1 immunofluorescence and a 'primed' microglial morphology (Fig. 2e, f, Supplementary Fig. 5a–c), also observed in aged hippocampi[16] (Supplementary Fig. 5d, e), whereas cortical microglia did not show an activated phenotype in AAV-IFNβ mice (Supplementary Fig. 5f–h).

**IFN-I impairs brain function in a microglia-dependent manner.** To directly test the impact of chronically increased IFN-I expression at the CP of young and middle-aged mice on the brain function, we administered AAV-IFNβ and AAV-CTRL viruses to wild-type mice and 4 weeks later we evaluated their hippocampus-dependent spatial learning and memory abilities, using the Radial Arm Water Maze (RAWM) and Novel Location Recognition (NLR) tests, and quantified synaptic puncta in their hippocampi (Fig. 3a). Interestingly, IFN-β overexpression had no effect on cognitive performance of young mice in either task (Supplementary Fig. 6a–e). However, middle-aged mice infected with AAV-IFNβ virus failed to identify novel location in the NLR test (Fig. 3b), committed more errors and took a significantly

longer time to locate the target platform in the RAWM task, relative to mice infected with AAV-CTRL (Fig. 3c, d). No differences in locomotive abilities between the age-matched groups were detected (Supplementary Fig. 6f, g). Further, we evaluated the number of presynaptic puncta in the CA3 and CA1 regions of the hippocampus, a measure strongly associated with performance in cognitive tasks, and decreased in ageing[17, 18]. We quantified VgluT2 and synaptophysin staining in CA3 and CA1 and found that numbers of synaptic puncta labelled by these molecules were decreased in the brains of IFN-β-overexpressing middle-aged mice (Fig. 3e–h).

To test whether the negative effect of IFN-I on the brain function is mediated by microglia, we created a mouse model of tamoxifen-inducible deletion of IFNAR in microglia, mic-IFNAR-KO (IFNAR^flox/flox^Cx3cr1^ERT2-Cre−/+^) mice. mic-IFNAR-CTRL (IFNAR^flox/flox^Cx3cr1^ERT2-Cre−/−^) littermates with normal IFNAR expression served as controls. We infected mic-IFNAR-KO and mic-IFNAR-CTRL mice with AAV-IFN-β or AAV-CTRL viruses (Fig. 3i). RNA-Seq of microglia isolated from whole brains of these animals revealed that mic-IFNAR-KO microglia were unresponsive to IFN-I overexpressed from the choroid plexus (Supplementary Dataset 4, Supplementary Fig. 7a), while the response of the whole hippocampus remained unchanged also in mic-IFNAR-CTRL animals (Supplementary Fig. 7b).

Next, we infected middle-aged mic-IFNAR-KO and mic-IFNAR-CTRL mice with AAV-IFN-β or AAV-CTRL viruses and tested their cognitive ability in the NLR test (Fig. 3j). Interestingly, the AAV-IFN-β-infected mic-IFNAR-KO showed a significantly increased preference to displaced object, whereas no preference was observed in AAV-IFN-β-infected mice with normal IFNAR expression, indicating that deletion of IFNAR in microglia was sufficient to prevent IFN-β-mediated cognitive loss. Quantification of synaptic VgluT2 and Synaptophysin immunostaining in CA1 and CA3 revealed that mic-IFNAR-KO mice were protected from IFN-β-induced loss of presynaptic puncta (Fig. 3k, l, Supplementary Fig. 7c).

Notably, in our previous study we showed that blockade of IFN-I signalling in the CNS of aged mice was sufficient to restore their cognitive ability[12]. In addition, we observed that 22-month-old mice lacking the IFN-I receptor (IFNAR-KO), or the key IFN-I-dependent transcription factor IRF-7 (IRF-7-KO) largely retained memory function during ageing as compared to age-matched wild-type controls, which showed reduced preference for a novel object in the NLR test (Supplementary Fig. 8a), further indicating a negative effect of IFN-I on brain function in old age. Collectively, these results suggest that the negative impact of chronically elevated IFN-I in the ageing brain is mediated to a large extent via its effect on microglia.

**Downstream effectors of microglial IFN-I signalling.** A thorough analysis of the RNA-Seq results revealed that the ageing brain microenvironment induced microglial expression of *B2M* and *C4b*, encoding B2M and C4 proteins, which have been linked with loss of cognitive ability and increased synaptic pruning, often accompanying brain pathology[19–22]. Analysis of the whole-brain microglial gene expression profile, hippocampal brain sections, and quantification of soluble B2M protein in the CSF of IFN-I-overexpressing middle-aged mice revealed that IFN-β promoted B2M production (Supplementary Fig. 9a–e). These results strongly suggest that microglia might be the source of B2M, previously observed in the ageing brain[19]. Similarly, analysis of whole-brain microglia for *C4b* at RNA level and staining of the hippocampal brain slices for C4 revealed IFN-I-mediated induction of C4 expression (Supplementary

Fig. 9f–i). Analysis of microglia of mic-IFNAR-CTRL and mic-IFNAR-KO infected with AAV-IFN-β or AAV-CTRL revealed a significant correlation between mRNA expression levels of IFN-I-dependent *Irf7* and those of *B2m* and *C4b* (Supplementary Dataset 4, Supplementary Fig. 9j, k). Moreover, levels of *B2m* and *C4b* increased upon activation of the IFNAR

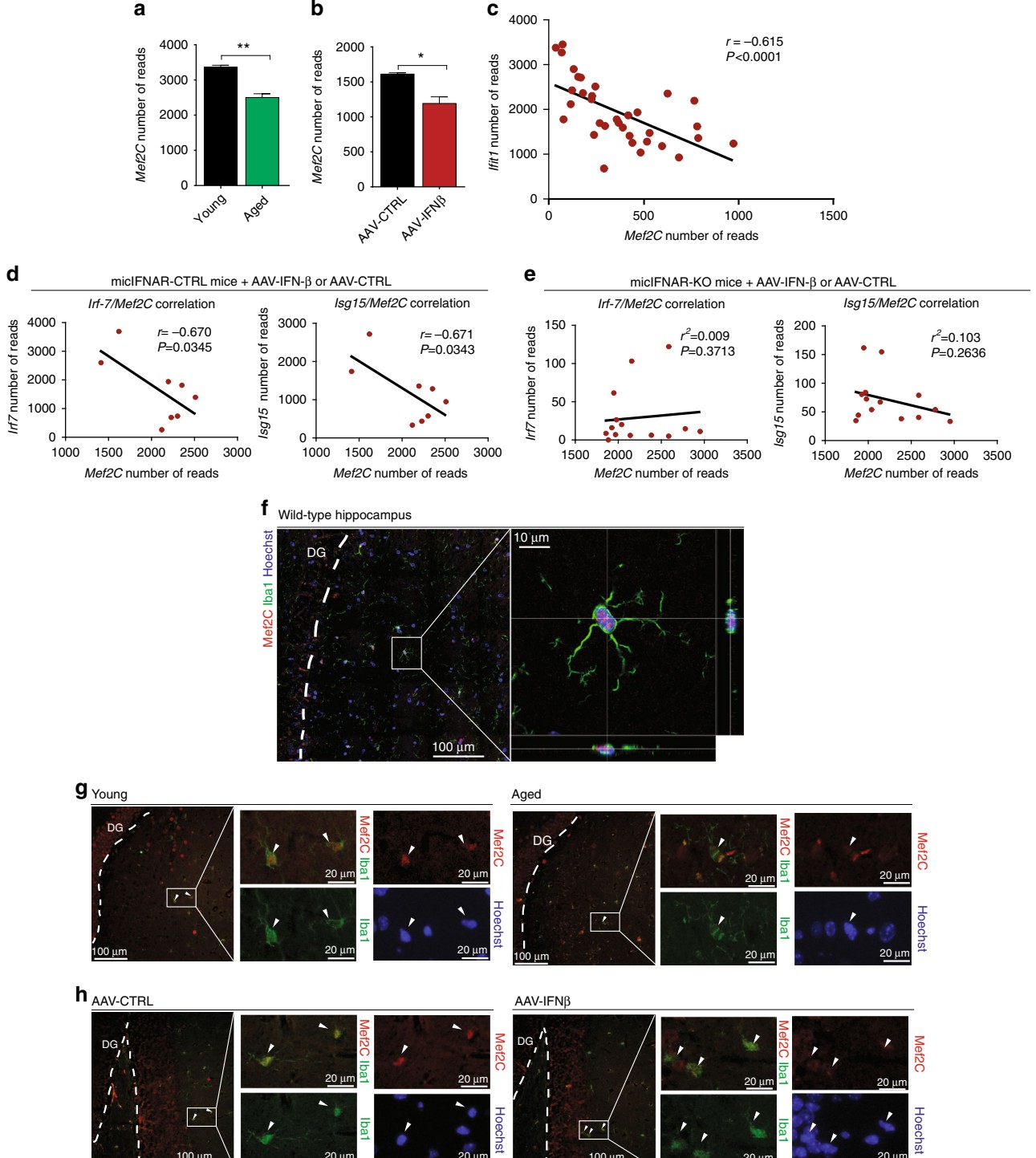

**Fig. 4** Microglial expression of Mef2C is reduced upon IFN-I overexpression and ageing. **a**, **b** mRNA expression levels of Mef2C (number of reads) in microglia of young and aged (**a**) and young AAV-IFNβ- and AAV-CTRL-infected mice (**b**). **\*\*** $P < 0.01$, **\*** $P < 0,05$; unpaired $t$-test, $n = 5$ per group. **c** Correlation between expression levels of *Mef2C* and IFN-I-dependent *Ifit1* in microglia of non-manipulated young and aged mice, young mice infected with AAV-IFNβ or AAV-CTRL and aged mice injected with anti-IFNAR antibody (data combined from Supplementary Datasets 1-3; Pearson $R^2 = 0.378$, $P < 0.0001$, $n = 34$). **d**, **e** Correlation analysis between expression levels of IFN-I-dependent *Irf7* and *Isg15*, and *Mef2C* reveals negative relationship in mic-IFNAR-CTRL microglia ($n = 8$) (**d**) but not in micIFNAR-KO microglia ($n = 13$) (**e**) suggesting direct effect of IFN-I on microglial expression levels of *Mef2C*. **f** Confocal images of nuclear (Hoechst nuclear staining in *blue*) Mef2C (*red*) expression in Iba1+ microglia (*green*) from adult wild-type mouse using orthogonal projections of confocal $z$-stacks. **g**, **h** Representative pictures of Mef2C staining (*red*), Iba1+ (*green*) microglia and Hoechst nuclear staining (*blue*) in hippocampal sections of non-manipulated young and aged mice (**g**) and young mice infected with AAV-IFNβ or AAV-CTRL (**h**)

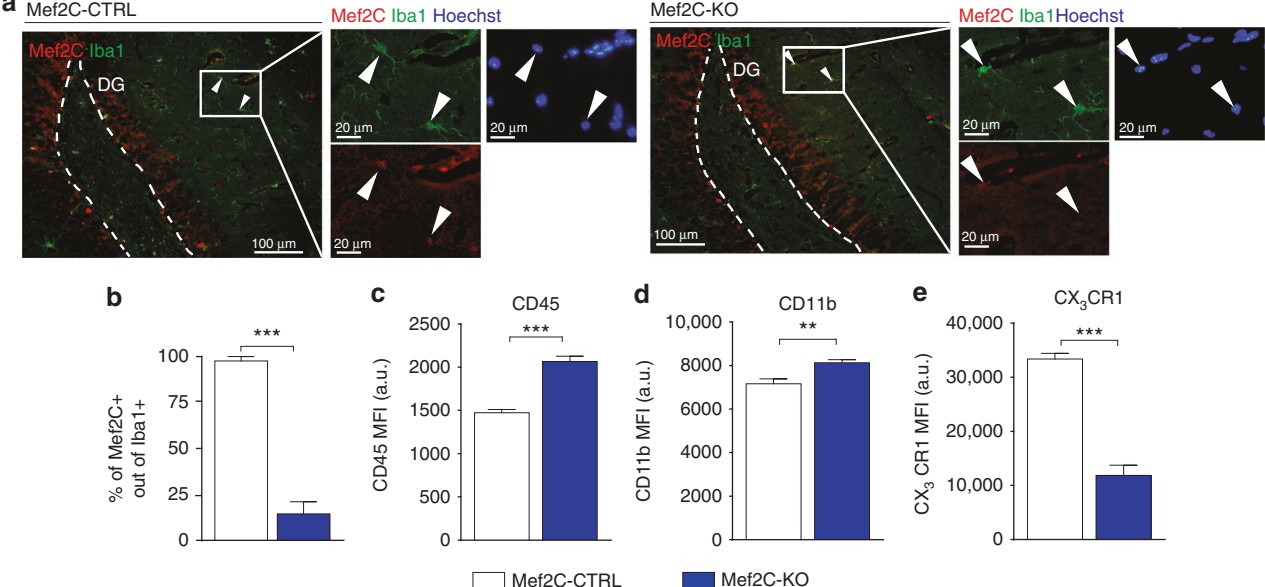

**Fig. 5** Loss of Mef2C alters microglial immune state. **a**, **b** Representative images (**a**) and quantification (**b**) of Mef2C (*red*) and Iba1 (*green*) staining in Mef2C-KO and Mef2C-CTRL mice 2 weeks following tamoxifen-induced recombination. **c–e** Mean fluorescent intensity of CD45 (**c**), CD11b (**d**) and CX3CR1 (**e**), in Mef2C-KO and Mef2C-CTRL microglia obtained by flow cytometry analysis. In all panels **$P < 0.01$, ***$P < 0.001$; unpaired $t$-test, $n = 8$–10 per group, the results are representative of four independent experiments

receptor on microglia (Supplementary Dataset 4, Supplementary Fig. 9l, m).

**Microglial Mef2C is downregulated upon chronic IFNβ**. We next attempted to identify IFN-I-dependent regulatory factors involved in shaping microglial ageing phenotype. Myocyte enhancer factor, Mef2C, is a transcription factor regulating various processes in multiple cells types, including immune cells and neurons[23, 24], and was previously proposed to regulate the microglial chromatin landscape[2]. Human GWAS studies linked mutations in Mef2C to ageing-associated late-onset Alzheimer's disease[25, 26]. RNA-Seq analysis of whole-brain microglia isolated from non-manipulated young and aged animals and young AAV-infected mice showed significant reduction of the *Mef2C* gene in ageing and upon chronic exposure to IFN-β (Fig. 4a, b, Supplementary Dataset 1 and 3). Furthermore, we compared microglial expression of *Mef2C* and the IFN-I-dependent *Ifit1* gene in sequencing data from young AAV-infected mice, non-manipulated young and aged mice, and aged animals intracerebroventricularly injected with anti-IFNAR neutralizing antibody (data combined from Supplementary Datasets 1–3). This analysis suggested that IFN-I signalling is negatively correlated with Mef2C expression (Fig. 4c, $r = -0.615$). In addition, analysis of RNA-Seq data of microglia from mic-IFNAR-CTRL and -KO mice confirmed the negative relation between expression levels of *Mef2C* and IFN-I-dependent genes *Irf7* and *Isg15* ($r = -0.670$ and $-0.671$ respectively), but only in mic-IFNAR-CTRL mice, suggesting that downregulation of Mef2C is dependent on a direct effect of IFN-I on microglia (Fig. 4d and e). To verify these findings at the protein level, we first confirmed nuclear localization of Mef2C transcription factor in microglia of wild-type mice using confocal imaging and orthogonal projection analysis (Fig. 4f). Immunohistochemical staining of hippocampal sections confirmed reduced levels of nuclear Mef2C in Iba1+ microglia in ageing and in AAV-IFNβ, compared to their controls (Fig. 4g, h, Supplementary Fig. 10a, b).

**Loss of Mef2c alters the microglial immune state**. To substantiate the role of Mef2C in the activity of microglia in vivo, we

created a mouse model of inducible Mef2C deficiency in microglia, by crossing Mef2C$^{loxP/loxP}$ with Cx3cr1$^{ERT2-Cre}$ mice. Microglia of adult Mef2C$^{loxP/loxP}$Cx3cr1$^{ERT2-Cre+/-}$ (Mef2C-KO) and Mef2C$^{loxP/loxP}$Cx3cr1$^{ERT2-Cre-/-}$ (Mef2C-CTRL) mice were analysed 2 weeks following tamoxifen-induced *Mef2C* inactivation. Immunohistochemical analysis revealed substantial elimination of Mef2C expression in Iba1+ microglia, but not in Iba1− brain cells in Mef2C-KO mice (Fig. 5a, b). Interestingly, initial flow cytometry analysis of whole-brain microglia of Mef2C-KO and Mef2C-CTRL mice revealed upregulation of CD45 and CD11b and concomitant reduction of CX3CR1 in microglia of Mef2C-KO mice when compared to Mef2C-CTRL mice (Fig. 5c–e, Supplementary Fig. 11a). The reduction of the CX3CR1 in the Mef2C-KO microglia, prompted us to test whether these mice would show behavioural changes, as loss of CX3CR1 was shown to affect cognitive performance[27]. To this end we tested Mef2C-KO and Mef2C-CTRL mice using a battery of behavioural tests for assessment of hippocampal-dependent spatial learning, working and long-term memory, social preference and anxiety. We did not observe any behavioural deficit that could be attributed to microglial Mef2C deficiency (Supplementary Fig. 11b–e). In addition, RNA-Seq of microglia of Mef2C-KO and Mef2C-CTRL mice did not reveal any strong transcriptional signature (Supplementary Dataset 5).

**Mef2C buffers microglial responses to immune stimuli**. Because the loss of microglial Mef2C was linked to downregulation of CX3CR1, one of the microglial 'off' molecules critical for restraining responses to immune stimuli, we hypothesized that Mef2C might be involved in limiting microglial activation upon immune deviation. Accordingly, the consequences of its loss may not be apparent under homoeostasis, but rather manifested in presence of inflammatory conditions, such as those that prevail in the ageing brain[28, 29]. To test this hypothesis, we delivered TNF, a pro-inflammatory cytokine associated with brain ageing[28], into the CSF of young Mef2C-KO and Mef2C-CTRL mice, and assessed the microglial response to this immune stimulus, and the effect on mice behaviour (Fig. 6a).

In agreement with the literature, after 24 h, TNF-injected Mef2C-CTRL mice exhibited a reduced preference for social interaction[30]; however, this effect was markedly enhanced in Mef2C-KO animals (Fig. 6b, c), suggesting their increased sensitivity to pro-inflammatory stimulation.

Immunohistochemical analysis of brain sections of TNF (or PBS)-injected mice 24 h following the injection revealed increased Iba-1 staining intensity in TNF-treated Mef2C-KO mice, when compared to TNF-injected Mef2C-CTRL mice (Fig. 6d and e). Furthermore, flow cytometry analysis revealed that upon immune

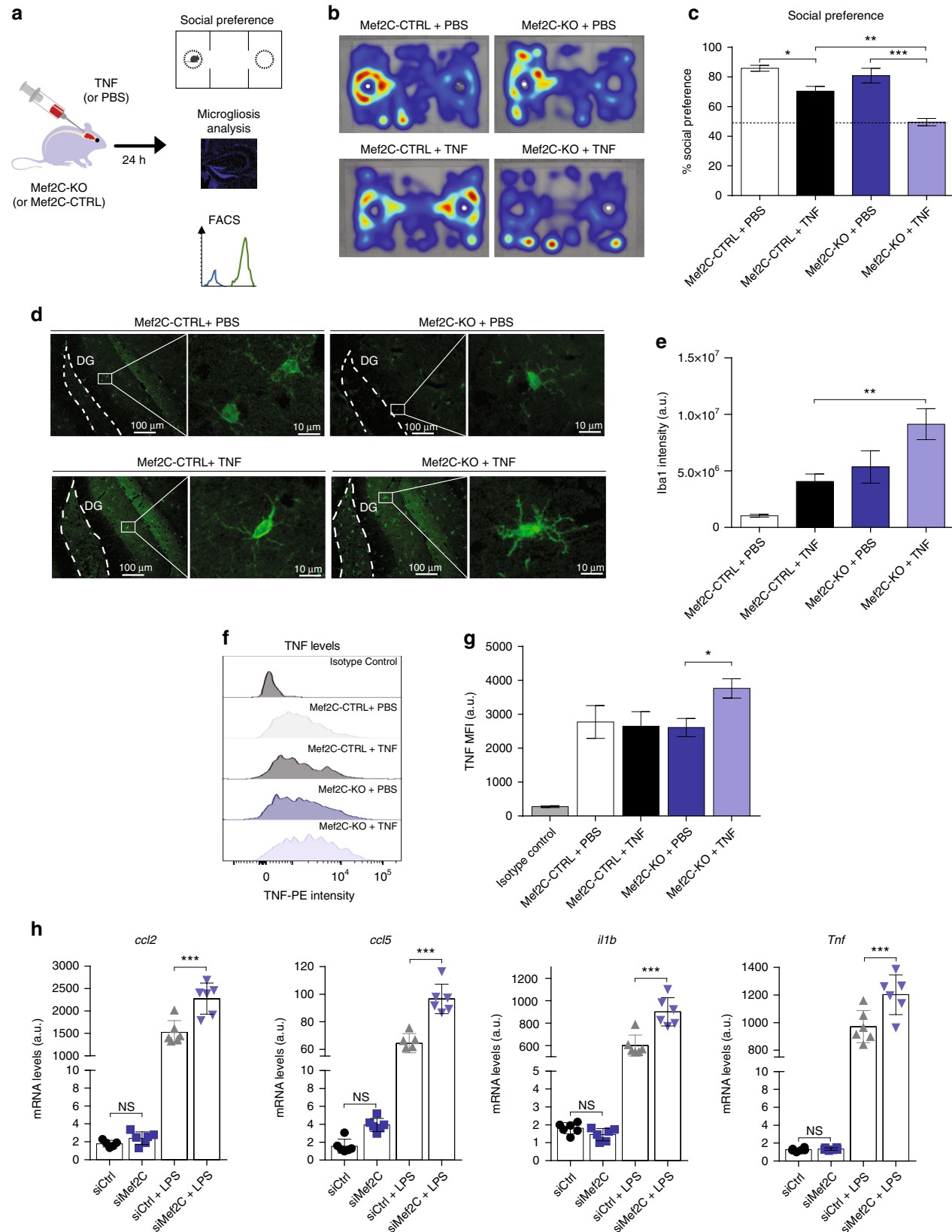

activation, Mef2C-KO microglia produced significantly higher levels of pro-inflammatory TNF than Mef2C-CTRL microglia (Fig. 6f, g). To test the transcriptional changes in microglia lacking Mef2C under immune challenge, we performed siRNA-mediated silencing of *Mef2C* in vitro cultured primary microglia, and stimulated the cells with lipopolysaccharide (LPS), a TLR4 agonist and a robust pro-inflammatory stimulus widely used in in vitro studies. RT-qPCR analysis revealed that effective and specific downregulation of Mef2C (Supplementary Fig. 12a) combined with LPS stimulation resulted in increased expression of pro-inflammatory chemokines (*Ccl2* and *Ccl5*) and cytokines (*Il-1b* and *Tnf*) (Fig. 6h), which are also overexpressed in aged microglia (Supplementary Fig. 2b). Altogether, these results suggest that Mef2C confers microglial resilience to immune challenge, and thereby promotes homoeostasis under pro-inflammatory conditions, which are present in the ageing brain[28, 29].

## Discussion

In the present study, we found that the transcriptional signature of aged brain microglia is largely determined by IFN-I, derived from the choroid plexus, and present in the ageing CNS milieu. In addition, chronic elevation of IFN-I signalling negatively affects brain function in adult mice in a microglia-dependent manner. Specifically, IFN-I-dependent transcriptional signature negatively correlated with microglial expression of the transcription factor, Mef2C. Using mice with genetic Mef2C deletion in microglia revealed the role of Mef2C in conferring resilience to pro-inflammatory stimuli, with impact on mice behaviour (Supplementary Fig. 13).

Microglial phenotype is largely affected by the cytokine milieu of the brain. Under pathological conditions, activities of multiple cytokines, including IFN-I, were reported to have various and even opposing effects on the brain, depending on the context[31–33]. Thus, for example IFN-I signalling was previously associated with attenuation of neuroinflammation[32, 34–36] and protection from neurodegeneration[37]. In contrast, chronically elevated IFN-I activity underlies or contributes to the pathology of various human CNS diseases, and in animal models, persistent expression or excessive activity of IFN-I in the CNS was previously linked to pathological changes, including ageing[12], neurodegeneration and microgliosis[12, 38–41]. Here, using multiple approaches, which include blockade of cerebral IFN-I signalling in aged mice, overexpression of IFN-I in the CNS of adult mice, and deletion of the IFN-I receptor in microglia, we provide evidence that the activity of aged microglia is largely and directly induced by IFN-I in the ageing brain milieu, and has a negative impact on cognitive ability. Similarly, microglia were proposed to mediate the effects of IFN-I on the brain in cases of viral infection[22, 33] and in an autoimmune disease[32].

Among the genes expressed by microglia upon exposure to IFN-I we identified MHC-I-related *B2m*, and the complement component *C4b*[19, 42], factors previously linked to cognitive impairment and ageing. Both MHC-I and complement factors were implicated in the synaptic pruning process, essential for brain development[7, 43], but detrimental if occurring in excess in the adult CNS[20, 42]. In addition, soluble B2M protein was reported to cause ageing-like cognitive decline and inhibit adult neurogenesis[19], and a recent study showed that cognitive impairment following cerebral viral infection is associated with microglia- and complement-dependent synaptic loss[22]. Altogether, IFN-I-dependent activity of microglia involves elevated production of factors associated with cognitive impairment in ageing.

Maintenance of CNS function is dependent on immune-restraining 'off' factors, which ensure risk-free microglial immune responses, and include the continuous presence of TGF-β in the soluble milieu of the CNS, interaction with neurons through certain microglial receptors (CX₃CR1, CD200), and expression of intracellular transcription regulators (e.g., MafB)[1, 9–11]. Loss of such factors and the consequences it inflicts on brain function were previously reported[27, 44, 45]. Thus, for example, treatment with recombinant CX₃CR1 ligand (CX₃CL1) reversed the age-related decrease in neurogenesis in rats[44].

Here we identify Mef2C as an additional 'off' factor, restricting microglial response to immune stimuli. Using tamoxifen-inducible CX3CR1^ERT2-Cre-based genetic manipulation, we observed that deletion of Mef2C increased the microglial response to the inflammatory CNS milieu, a common feature of brain ageing and neurodegenerative conditions[28, 29]. Almost paradoxically, in an aged brain, where such conditions prevail and microglial restraints are most urgently needed, we observed that microglial Mef2C was downregulated, possibly as an outcome of chronic IFN-I presence in the brain. Our data suggest that loss of Mef2C may be involved in 'priming', a microglial state of increased sensitivity to immune stimuli and a hallmark of ageing[28, 29, 46]. Furthermore, human GWAS studies linked mutations in Mef2C to ageing-associated late onset Alzheimer's disease[25, 26]. On the basis of our results, it is possible that genetic or ageing-induced loss of Mef2C function in microglia can contribute to the pro-inflammatory milieu of the brain in Alzheimer's disease or ageing, thereby exacerbating cognitive loss and disease pathology[28]. Notably, however, in certain pathologies (e.g., Amyotrophic Lateral Sclerosis[47] and specific stages of Alzheimer's disease[48]) transient reduction of such restraining mechanisms was linked to beneficial outcome, highlighting the fine balance of regulation and suggesting that the role of microglial "off" molecules, including Mef2C, may be disadvantageous in specific pathological contexts.

In summary, we demonstrate that microglial activity in ageing is largely shaped by IFN-I, whose levels increase in the brain in old age. Our data suggests that ageing-related loss of cognitive ability is, at least in part, dependent on this microglial signature. Among several possible mechanisms linking microglial IFN-I signature and ageing-related cognitive loss, our results identify IFN-I-dependent downregulation of Mef2C, a microglial "off" factor, which we find to be involved in restraining microglial responses to immune stimuli. Accordingly, it implies that the

**Fig. 6** Mef2C deficiency is associated with increased pro-inflammatory response to immune stimuli. **a** Scheme illustrating the experimental design to determine the effects of immune challenge in Mef2C-KO mice. **b**, **c** Social preference assessed in Mef2C-KO and Mef2C-CTRL mice 24 h after intracerebroventricular injection of TNF or PBS; representative heatmaps (*white dot* indicates the stranger mouse location) (**b**) and percentage of social preference (**c**) (representative of two independent experiments, *n* = 3 per group). **d**, **e** Representative micrographs (**d**) and quantification (**e**) of Iba-1 staining in microglia in the hippocampus of Mef2C-KO and Mef2C-CTRL mice 24 h after intracerebroventricular injection of PBS or TNF. (*n* = 5 per group). **f**, **g** Representative histograms (**f**) and quantification of mean fluorescence intensity (**g**) of TNF staining in Mef2C-KO and Mef2C-CTRL mouse microglia 24 h after intracerebroventricular TNF or PBS administration. (*n* = 5 per group) **h** mRNA expression levels of pro-inflammatory *ccl2, ccl5, il1b, Tnf* in primary microglial cultures following siRNA-mediated Mef2C depletion and treatment with LPS (*n* = 4 per group). In all panels *$P < 0.05$, **$P < 0.01$, ***$P < 0.001$; one-way ANOVA with Newmann–Kleus post hoc test, the results are representative of 2–3 independent experiments

activity of Mef2C becomes critical under pathological conditions and in ageing, when levels of inflammatory cytokines are elevated. Together, our findings may shed light on several mechanisms present in other ageing-related brain conditions and neurological diseases associated with elevated IFN-I in the CNS.

## Methods

**Animals.** Wild-type aged mice (18–22 months old; Weizmann Institute of Science colony), 10 months old, 8–10 weeks old and neonatal (P0–P1) C57BL/6 J mice (Envigo, Israel) were used. *Mef2ctm1Jjs/J* (Mef2C$^{loxP}$ mice)[49] were purchased from The Jackson Laboratory (stock no. 025556). Cx3cr1$^{ERT2-Cre}$ mice[50] were a kind gift of Prof. Steffan Jung, Weizmann Institute of Science. IFRNAR-KO, IRF-7-KO and IFNAR$^{flox}$ mice[32] were a kind gift of Prof. Marco Prinz. Mef2C$^{loxP}$ mice carrying loxP-flanked exon of *Mef2c* or IFNAR$^{flox}$ mice with flox-flanked exon of *Ifnar1* were crossed with Cx3cr1$^{ERT2-Cre}$ transgenic mice expressing ERT2-Cre recombinase under CX$_3$CR1 promoter. For induction of recombination, 5-week-old to 7-week-old Mef2C$^{loxP/loxP}$ Cx3cr1$^{ERT2-Cre+/-}$ (Mef2C-KO) and Mef2C$^{loxP/loxP}$ Cx3cr1$^{ERT2-Cre-/-}$ or Mef2C$^{-/-}$ Cx3cr1$^{CreER+/+}$ (Mef2C-CTRL) or IFNAR$^{fl/fl}$ Cx3cr1$^{ERT2-Cre+/-}$ (micIFNAR-KO) and IFNAR$^{fl/fl}$ Cx3cr1$^{ERT2-Cre-/-}$ (micIFNAR-CTRL) mice were gavaged with 4 mg tamoxifen (TAM, Sigma) dissolved in 200 µl corn oil (Sigma) five times, every other day in accordance with previous reports[50]. Only male mice were used throughout the study. All mice were housed and bred under specific pathogen-free conditions, in a 12 h light–dark cycle. All behavioural tests were conducted during the dark hours, in dimly lit room. Permission to perform the animal experiments was granted by the Weizmann Institute of Science Institutional Animal Care and Use Committee (IACUC).

**Radial arm water maze.** The radial arm water maze (RAWM) test was performed during dark hours, in a dimly lighted room, as previously described[51]. Briefly, the goal arm location (containing a platform submerged 1.5 cm below the water surface) remained constant for a given mouse, whereas the start arm was changed during each trial. Within the testing room, only distal visual shape and object cues were available to aid location of the platform. On day 1, mice were trained for 15 trials, with alternating visible and hidden platform. On day 2, mice were trained for 15 trials with the hidden platform only. Entry into an incorrect arm, or remaining for more than 15 s in the same incorrect arm, or the central area, was scored as an error. The number of errors was determined for each trial. The raw data were analysed as mean of errors for training blocks, each spanning three consecutive trials. Swimming speed was measured using EthoVision automated tracking system (Noldus).

**Novel location recognition test.** Novel location recognition test (NLR) was performed during dark hours, in a dimly lighted room, as previously described[52]. Briefly, mice were placed in a grey, square box (50 × 50 × 50 cm) with visual cues on the walls. On the training day, mice were given four sessions of 6 min; during the first session, mice were allowed to explore the arena without objects (habituation), and in the following three trials, two objects of different colour, shape and texture were present in defined locations in the box (training, day 1). After 24 h, mice were returned to the arena, in which one of the objects was placed in a new location (testing day, day 2). Time spent exploring each object on each day was manually scored using EthoVision tracking system (Noldus), and percentage preference for the displaced object was calculated for each animal, for each day, by dividing the time spent exploring the displaced object by the total exploration time of both objects and multiplying the result by 100%, according to the formula: Percentage preference = ((displaced object exploration time)/(displaced object exploration time + non-displaced object exploration time)) ×100%. The result of the calculation was approximately 50% on day 1 (training) and 60–90% on day 2 for mice with normal memory.

**Open field.** Open field for assessment of mobility and anxiety was performed according to previous reports[53]. Mice were allowed to explore a grey square box (50 × 50 cm) for 10 min. Velocity and time spent in the centre arena (25 × 25 cm) were acquired using EthoVision tracking system (Noldus).

**Y-maze.** Y-maze to assess working memory by measuring spontaneous alternation was performed as previously described[54]. Briefly, the subject mouse was placed in the middle of a three-armed Y-shaped maze, and was allowed to freely explore the maze for 5 min. During this time the order of entries to the arms was recorded using EthoVision tracking system (Noldus). All possible triplets of consecutive entries were scored: entry to three different arms was scored as "1", entry to the same arm within three consecutive entries was scored as "0". Percentage alternation was calculated for each animal by dividing the sum of acquired points by a sum of possible triplets and multiplied by 100%.

**Social preference.** Social preference test for assessment of social behaviour was performed as previously described[55]. Briefly, mice were allowed to explore three-chamber arena for 10 min. After this time, mice were separated in the middle

chamber and small wired cages were placed in the side chambers: one containing a stranger mouse of the same phenotype and age and one empty. Mice were allowed to explore the arena with wired cages for 10 min and exploration time of each cage was scored using EthoVision tracking system (Noldus). Percentage preference for the cage containing a stranger mouse over empty cage was calculated for each animal, according to the formula (time spend exploring stranger mouse cage/(time spend exploring stranger mouse cage + time spent exploring the empty cage)) ×100%. The arena and wired cages were wiped with 10% ethanol and the location of the stranger mouse was altered between the tested animals.

**Intracerebroventricular injections.** Intracerebroventricular injections were performed as previously described[56]. In brief, mice were anesthetised with isoflurane and placed on a warm pad in a stereotactic device (World Precision Instruments) and injected subcutaneously with painkiller (Carprofen 5 mg/kg). Skin over head was incised, connective tissue over the scull removed with H$_2$O$_2$. Injection site was determined 0.4 mm posterior to the bregma, 1.0 mm lateral to the midline. An opening in the scull was drilled and blunt 30 gauge needle attached to a syringe (Hamilton) was inserted to 2.0 mm in depth from the brain surface. After 3 min delay time, neutralizing antibody to IFNAR1 (mouse anti-mouse IFNR1 antibody clone MAR1-5A3, eBioscience) (10 µg) or mouse IgG1 κ isotype control (clone MG1-45, BioLegend) (10 µg) (Fig. 1), or TNF (50 ng in 5 µl PBS) or 5 µl PBS (Fig. 6) were injected at 0,5 µl/min speed using UMP3 UltraMicroPump (World Precision Instruments). After the desired volume was injected, and another 3 min delay time, the needle was removed, the skin sutured and the mouse injected subcutaneously with 1 ml saline for rehydration and placed in a recovery cage under warm lamp.

**AAV preparation and injection.** Low passage HEK293T (ATCC, catalogue no. CRL-3216) were maintained at 37 °C with 5% CO$_2$ in Dulbecco's modified Eagle's medium supplemented with 10% fetal bovine serum. Cell line was tested for mycoplasma contamination using EZ-PCR mycoplasma test kit (Biological Industries, catalogue no. 20-700-20). To produce AAV-IFNβ and AAV-Ctrl, a triple co-transfection procedure was used to introduce an AAV vector plasmid (pAAV-CAG-IFNβ-IRES-GFP or pAAV-CAG-IRES-GFP, respectively) together with pXR1 AAV helper plasmid carrying AAV rep and cap genes and pXX6-80 Ad helper plasmid at a 1:1:1 molar ratio[57]. HEK293T cells were transfected using poly-ethylenimine (linear; molecular weight 25,000) (Poly- sciences, Inc., Warrington, PA), and medium was replaced at 12 h post transfection. 72 h post transfection conditioned medium was collected, cells were harvested and subjected to 3 rounds of freeze-thawing, and then digested with 100 U/ml Benzonase (EMD Millipore, Billerica, MA) at 37 °C for 1 h. Viral vectors were purified by iodixanol (Sigma, Israel) gradient ultracentrifugation[58] followed by further concentration using Amicon ultra-15 100 K (100,000-molecular-weight cutoff, Merck Millipore) and washed with phosphate-buffered saline (PBS). Final concentration of AAV1 particles was 5.73E + 8/ µl for AAV-IFNβ and 3.33E + 8/ µl for AAV-CTRL. Viruses were injected intracerebroventricularly (0.4 mm posterior to the bregma, 1.0 mm lateral to the midline and 2.0 mm in depth from the brain surface), as described[56] in a quantity of 1E + 9 vg/animal in volume of 10 µl.

**Cerebrospinal fluid collection and ELISA analysis.** Cerebrospinal fluid (CSF) was collected by the cisterna magna puncture technique[59]. In brief, mice were anaesthetized with intraperitoneal injection of ketamine (100 mg/kg) and xylazine (10 mg/kg) and placed on a stereotactic instrument so that the head formed a 135° angle with the body. A sagittal incision of the skin was made inferior to the occiput and the subcutaneous tissue and muscles were separated, and a capillary was inserted into the cisterna magna through the dura matter lateral to the arteria dorsalis spinalis. Approximately 10 µl of CSF could be aspirated from an individual mouse. Levels of B2M in CSF samples were measured by ELISA (Cloud-Clone Corp, catalogue no. SEA260Mu) according to the manufacturer's protocol.

**Sorting and flow cytometry analysis of microglia.** Mice were anaesthetized by intraperitoneal injection of ketamine (100 mg/kg) and xylazine (10 mg/kg) and perfused with PBS to the left heart ventricle prior to tissue collection. Next, brain was isolated, choroid plexi and meninges were removed. Remaining brain tissue was mechanically dissociated using a software-controlled sealed homogenization system (Dispomix; http://www.biocellisolation.com). Cell suspension filtered through an 80-µm nylon mesh, centrifuged over a 40% Percoll gradient (GE Healthcare) for 20 min at 2250 r.p.m. with minimal acceleration and deceleration, washed once and blocked with Fc-block CD16/32 (1:100, BD Biosciences). Next, samples were stained using PerCP-Cy5.5-conjugated anti- CD11b (1:200, Biolegend M1/70), Brilliant Violet 421-conjugated anti-CD45 (1:150, Biolegend 30-F11), PE-conjugated anti-CX$_3$CR1 (1:150, Biolegend SA011F11) and APC-conjugated anti- Ly6C antibodies (1:150, Biolegend HK1.4). CD11b$^+$CD45$^{int}$Ly6C$^-$ microglia (3500 cells) were sorted using SORP-Aria sorter (BD Biosciences) into 80 µl of Lysis/Binding buffer (Invitrogen). Results were analysed using FACSDiva and FlowJo softwares. Unstained control was used to identify the population of interest and to exclude others. TNF intracellular staining was performed as previously described[60], using BD Cytofix/Cytoperm TM Plus kit (BD Biosciences; catalogue

no. 554715) and PE-conjugated anti- TNF-α (1:50, Biolegend MP6-XT22) and isotype control (1:50, PE-conjugated rat IgG1, κ).

**RNA purification and library preparation**. Cells were harvested into Lysis/Binding buffer (Invitrogen). mRNA was captured with 12 μl of Dynabeads Oligo (dT) (Life Technologies), washed, and eluted at 70 °C with 10 μl of 10 mM Tris-Cl (pH 7.5). RNA-seq was performed as previously described[61] and DNA libraries were sequenced on an Illumina NextSeq 500 or HiSeq with an average of 4 million aligned reads per sample.

**RNA-Seq processing and analysis**. We aligned the RNA-seq reads to the mouse reference genome (NCBI 37, mm9) using TopHat v2.0.13 with default parameters[62]. Duplicate reads were filtered if they aligned to the same base and had identical UMIs. Expression levels were calculated and normalized for each sample to the total number of reads using HOMER software (http://homer.salk.edu) with the command "analyzeRepeats.pl rna mm9 -d [sample files] -count 3utr -condenseGenes"[63].

For the analysis of RNA-seq in ageing, following anti-IFNAR treatment (Fig. 1) and following IFN-β overexpression (Fig. 2), we focused on genes with mean expression above noise (set at 128). We defined age-related genes based on a two-tailed $t$-test ($P < 0.05$) and whether they increased or decreased with ageing. We used equivalent thresholds for the interferon treatment and IFNAR blockade experiments with differentially-regulated genes identified based on two-tailed t-test ($P < 0.05$). We calculated overlap between either the increased genes (356) or the decreased genes (318) and up-/downregulated in AAV-IFN-β-expressing (248/198) or IFNAR (Day 2 = 216/199, Day 7 = 198/247, Day 14 = 318/200) genes using a hypergeometric distribution. For the RNA-seq analysis of siMef2C experiment, we focused on genes with mean expression above noise (set at 128), and differentially expressed (with 1.5-fold differential, log fold change = 0.585) between the means of the groups (two-tailed $t$-test, $P < 0.01$). For GO enrichment analysis we used Gene Ontology enRIchment anaLysis and visuaLizAtion tool (Gorilla; http://cbl-gorilla.cs.technion.ac.il/)[64, 65], to which gene symbols of significantly increased or decreased genes (as a target set) and a complete gene list (as a background set) were imported. Venn diagram of IFN type I (α and β), II (IFN-γ), and III-dependent genes was created using Interferome v. 2.01 (http://www.interferome.org/interferome/search/showSearch.jspx)[66] with default parameters. Heat-map was prepared using GENE-E (http://www.broadinstitute.org/cancer/software/GENE-E/).

**cDNA synthesis and quantitative real-time PCR analysis**. Cultured primary newborn mouse microglia, frozen hippocampi and spleens were dissociated mechanically in TRI reagent (MRC, Cincinnati, OH) and RNA was extracted and purified from the lysates using the RNeasy kit (Qiagen; catalogue no. 74104). RNA from CP and cultured CP epithelial cells was extracted and purified using RNA MicroPrep kit (Zymo Research; catalogue no. R1050). mRNA (1 μg) was converted into cDNA using a High Capacity cDNA Reverse Transcription Kit (Applied Biosystems; catalogue no. 4368814). mRNA expression levels of specific genes at the CP were assessed using fluorescence-based quantitative real-time PCR (RT-qPCR). Quantification reactions were performed in duplicate for each sample using the "delta-delta Ct" method. Peptidylprolyl isomerase A (*ppia*) and Hypoxanthine-guanine phosphoribosyltransferase (*hprt*) were chosen as reference (housekeeping) genes. RT-qPCR reactions were performed and analysed using StepOne software V2.2.2 (Applied Biosystems). The primer sequences are listed in Supplementary Table 1.

**Immunohistochemistry**. Paraffin-embedded 6 μm thick mouse brains coronal and sagittal sections were used for immunohistochemistry analyses. For immuno-fluorescent staining, the following primary antibodies were used: goat anti-Iba1 (polyclonal, 1:100, Abcam), rabbit anti-B2M (polyclonal, 1:100, Abcam), rabbit anti-human C4a/b (cross-reactive with murine C4; polyclonal, 1:100, Bioss), rabbit anti-GFP (polyclonal, 1:100, MBL), mouse anti-VgluT2 (8G9.2, 1:100, Abcam), rabbit anti-Synaptophysin (polyclonal, 1:400, Abcam), mouse anti-Mef2C (1H5, 1:100, Novus) and mouse anti-Claudin-1 (2H10D10, 1:100, Invitrogen) with use of Mouse on Mouse (M.O.M.) Basic Kit (Vector Labs; catalogue no. BMK-2202). Secondary antibodies included: Cy2/Cy3 -conjugated donkey anti-rabbit/goat/mouse antibodies (1:200; all from Jackson ImmunoResearch). The slides were exposed to Hoechst for nuclear staining (1:2000; Invitrogen Probes) for 30 seconds. Negative controls routinely used in immunostaining procedures included staining with secondary antibody alone, and staining without primary and without secondary antibody to exclude non-specific binding of the secondary antibody. Images were taken using confocal microscope (LSM880, Carl Zeiss, Inc.) or fluorescence microscope (Nikon Eclipse 80i), equipped with 20 × NA 0.50 and 40 × NA 0.75 objective lenses (Plan Fluor; Nikon) and a digital camera (DXM 1200 F; Nikon) using acquisition software (NIS-Elements, F3). Images were cropped and merged using Photoshop CS6 13.0 (Adobe), and were arranged using Illustrator CS5 15.1 (Adobe). For quantification of staining intensity (Iba1 and synaptophysin) total and background fluorescence was measured from the areas of interest using ImageJ software (NIH), and intensity of specific staining was calculated, as previously described[67]. For quantification of Mef2C+ microglia double-stained Mef2C+Iba-1+ and single-stained Iba1+ cells with distinct microglial

morphology cells were counted from five sections per mouse using ImageJ. Number of Mef2C+Iba-1+ cells was divided by the number of Iba-1+ cells and multiplied by 100%. For quantification of VgluT2 puncta Image Pro (Media Cybernetics) was used as previously described[68]. Background subtraction using Rolling ball was done in Fiji and cell/neurite tree morphology quantification was done using NeuroMath[69, 70].

**Primary newborn mouse microglial culture**. Microglial primary cultures were prepared as described in ref. [34]. Meninges and choroid plexi were removed under a dissecting microscope (Stemi DV4; Zeiss) from neonatal (P0–P1) C57Bl/6 J mice brains in Leibovitz-15 medium (Biological Industries). After trypsinization (0.5% trypsin, 10 min, 37 °C), the tissue was mechanically dissociated. Next, the brain glial cells were re-suspended in DMEM supplemented with 10% FCS, 1 mM L-glutamine, 1 mM sodium pyruvate, 100 U/ml penicillin, and 100 μg/ml streptomycin, and cultured at 37 °C, 5% $CO^2$ in 75-cm² Falcon tissue-culture flasks (BD Biosciences) pre-coated with poly-D-lysine (PDL) (10 μg/ml; Sigma-Aldricht). The medium was replaced after 24 h and every other day thereafter, for 14 days. Microglia were shaken off the primary mixed brain glial cell cultures (170 r.p.m., 37 °C, 6 h), re-suspended in microglial culture medium (RPMI-1640 medium (Sigma-Aldrich) supplemented with 10% FCS, 1 mM L-glutamine, 1 mM sodium pyruvate, 100 U/ml penicillin, and 100 μg/ml streptomycin, as well as mouse recombinant MCSF 10ng/ml (Peprotech) and 50ng/ml human recombinant TGFβ1 (Peprotech) to better mimic in vivo conditions[1] and seeded ($10^5$ cells/ml) in 24-well plates (1 ml/well; Corning) pretreated with poly-D-lysine.

**RNA interference and RNA purification**. Cultured primary newborn mouse microglia were transfected with 100 nM siRNA directed against Mef2C (ON-TARGETplus Mef2c siRNA, Dharmacon, catalogue number: L-002000-00-0005; target sequences of siMef2C duplexes: 5′-GAGGAUCACCGGAACGAAU-3′, 5′-UAGUAUGUCUCCUGGUGUA-3′, 5′-GAUAAUGGAUGAGCGUAAC-3′, 5′-CCAGAUCUCCGCGUUCUUA-3′) or non-targeting siRNA (Dharmacon, catalogue number: D-001810-10-05) with Lipofectamine 2000 (Invitrogen), according to the manufacturer's guidelines. Briefly, Lipofectamine was diluted in Opti-MEM I Reduced Serum Medium (Invitrogen) and incubated for 20 min at room temperature. Next, siRNA was diluted in Opti-MEMI Reduced Serum Medium (Invitrogen) and incubated for 5 min at room temperature. Subsequently, both dilutions (of siRNA and Lipofectamine) were mixed together and added to microglial cultures. After 5 h microglial medium was added. After 20 h cells were additionally treated with 100 ng/ml lipopolysaccharide (LPS; E. coli 055:B5, Sigma-Aldrich) and collected 4 h after (24 h from the start of transfection) using TRI reagent (MRC). RNA was purified from the lysates using the RNeasy kit (Qiagen; catalogue no. 74104).

**Primary culture of choroid plexus cells**. CP cultures were prepared as previously described[56]. Briefly, the CP tissue excised from PBS-perfused animals was dissociated in 0.25% trypsin by shaking (20′ at 37 °C) and pipetting, then centrifuged, washed and plated (~ 250,000 cells/well) in 24-well plates in culture medium for epithelial cells (DMEM/HAM's F12 (Invitrogen Corp)), supplemented with 10% Fetal Calf Serum (Sigma-Aldrich), 1 mM l-glutamine, 1 mM sodium pyruvate, 100 U/ml penicillin, 100 mg/ml streptomycin, 5 μg/ml insulin, 5ng/ml sodium selenite, 20 μM arabinofuranosyl cytidine (Ara-C) and 10 ng/ml EGF, at 37 °C, 5% $CO_2$. After 24 h, the medium was changed, and the cells were either left untreated, or incubated with medium supplemented with 25% or 50% of medium conditioned of HEK293T cells used to produce AAV-IFNβ or AAV-Ctrl viruses. After 24 h, cells were washed twice with PBS and RNA isolation was performed with RNA MicroPrep kit (Zymo Research; catalogue no. R1050), according to the manufacturer's protocol.

**Statistical analysis**. Results are presented as means ± s.e.m. The specific tests used to analyse each set of experiments are indicated in the figure legends. All experiments were repeated at least twice and representative graphs and pictures are presented. Data were analysed using a two-tailed Student's $t$-test to compare between two groups, one-way ANOVA was used to compare several groups, followed by the Newman–Keuls post-hoc procedure for pairwise comparison of groups after the null hypothesis was rejected ($P < 0.05$). Data from RAWM were analysed using two-way repeated-measures ANOVA, and Bonferroni post hoc procedure was used for follow-up pairwise comparison. Sample sizes were chosen with adequate statistical power based on the literature and past experience, and mice were allocated to experimental groups according to age and genotype. No specific randomization protocols were used. Investigators were blinded to the identity of the groups during experiments and outcome assessment. Statistical calculations were performed using GraphPad Prism software (GraphPad Software).

**Data availability**. The authors declare that all data supporting the findings of this study are available within the article and its Supplementary Information files. The RNA Sequencing data have been deposited in NCBI's Sequence Read Archive under the entry code GSE98401.

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

## Acknowledgements

We thank Prof Steffen Jung for the kind gift of the Cx3cr1$^{ERT2-Cre}$ mice, Margalit Azulay for assistance with animal care, Prof Alon Chen, Raaya Zwang, and Maya Groysman for help with preparation of viruses for preliminary experiments, Dr Ofer Yizhar and his group for sharing stereotactic injection facilities, Dr Shelley Schwarzbaum for manuscript editing, and Tal Bigdary for artwork. Research in the M.S. lab is supported by Advanced European Research Council grants (232835), and by the EU Seventh Framework Program HEALTH-2011 (279017); Israel Science Foundation (ISF)-research grant no. 991/16, ISF-Legacy Heritage Biomedical Science Partnership-research grant no. 1354/15, and Consolidated Anti-Aging Foundation (CAFF). M.S. holds the Maurice and Ilse Katz Professorial Chair in Neuroimmunology. Research in the I.A. lab is supported by the European Research Council (309788), the Israeli Science Foundation (1782/11), the BLUEPRINT FP7 consortium, the Ernest and Bonnie Beutler Research Program of Excellence in Genomic Medicine, a Minerva Stiftung research grant, the Israeli Ministry of Science, Technology and Space, the David and Fela Shapell Family Foundation, and the National Human Genome Research Institute Center for Excellence in Genome Science (1P50HG006193). I.A. is the incumbent of the Alan and Laraine Fischer Career Development Chair. M.P. is supported by the BMBF-funded competence network of multiple sclerosis (KKNMS), the Sobek-Stiftung, the DFG (SFB 992, SFB1140, Reinhart-Koselleck-Grant), the ERA-Net NEURON initiative "NEURO-IFN" and the Sonderlinie Hochschulmedizin, project "neuroinflammation in neurodegeneration". I.A. and M.P. are supported by the SFB/TRR167.

## Author contributions

A.D., O.M.-N., I.A. and M.S. conceived the project, A.D., O.M.-N., I.A., M.S. designed the experiments; A.D., O.M.-N., A.T.-K., S.B.-H., R.D.-S., N.R., K.B. A.K. (under supervision of M.S. and I.A.) and L.K.S. (under supervision of S.V.) performed the experiments; A.S., E.D. and D.R.W. analysed the sequencing data (under supervision of I.A.); A.T. under supervision of A.C. prepared the viral construct, A.T.-K. and O.S. prepared viruses; M.P. contributed transgenic mice and edited the manuscript, A.D., O.M.-N., M.S and I.A. wrote the manuscript; M.S. and I.A. supervised the project.

## Additional information

**Competing interests:** The authors declare no competing financial interests.

