## [Peer Review File · Nature Communications]

Reviewers' comments:

Reviewer #1 (Remarks to the Author):

This manuscript considers the functional overlap between age, IFN β (in the choroid plexus) and downregulation of a transcript factor, which the authors suggest is a key factor in modulating microglial activation. A good deal of data is presented, some of which is predictable based on the authors previous work, or not especially new. What is new, is the finding that Mef2C can exert a modulatory effect on microglia but this aspect of the work is under-developed and preliminary, and the impact of age and IFN β on its expression, the consequent microglial activation and neuronal function needs further work.

Specific major issues:

The abstract

1. makes reference to a microglial "homeostatic phenotype" but this is not properly defined anywhere in the text;
2. states that the 'aged microglial phenotype is imposed by the cerebrospinal fluid microenvironment' but this is not shown;
3. concludes that "detrimental aging-related activity of microglia to be largely imposed by the aged brain microenvironment, and identified Mef2C as a key transcription factor regulating microglial homeostatic function under immune challenge", but only the second part of this is actually demonstrated.

Lines 107-109: It is stated that IFN β confined to brain - it would be better to state that it is confined to the choroid plexus since there was no change in the hippocampus and, with respect to the persistent response at 7 months, this was evaluated only in CP so being specific in the first place is appropriate.

The implication is that IFN β stimulates microglial activation in the hippocampus but there is no evidence of increased IFN β in the brain tissue so the authors need to spell out how changes can be attributed to IFN β . With respect to Fig 1g, labelling indicates changes in hippocampus; do the data presented in Fig 1h related to the hippocampus (the whole hippocampus or a sub-field)? This should be stated. In this context, VgluT2 was assessed in area CA3; why was this area selected for analysis and what changes, if any were found in other sub-fields? No details are given about what area of the brain was used to prepare microglia (Fig 2g and 2k), nor what area was assessed for β 2M or C4a/b immunoreactivity (Fig 2i and 2m).

While the data indicate anti-IFNAR antibody treatment reduced age-related gene expression, it is odd that neither behaviour nor microglial activation were evaluated.

Mef2C becomes the focus of attention from Figure 4 and this shift is somewhat odd given that anti-IFNAR antibody did not significantly rescue the age-related change (although decreases with age and IFN β overexpression were observed). With respect to Fig 4, what brain area was used to evaluate Mef2C mRNA?

The impact of downregulating Mef2C was assessed in vitro and in vivo but the readouts were different for the 2 experiments and it would certainly strengthen the argument for a key modulatory role for Mef2C if at least some of the same readouts were assessed. It is not clear why spatial learning was not assessed in Mef2C KO mice.

Fig 6a,b: It is not acceptable to present SEM values or to undertake statistical analysis when data are

derived from only 2 replicates.

Reviewer #2 (Remarks to the Author):

Deczkowska, Matcovitch-Natan et al. present evidence for a CSF-microglia axis that potentially modulates microglial homeostasis during aging. Through a series of over-expression and antibody blocking experiments, they demonstrate that elevated IFN-B production in the brain is sufficient to drive changes in microglia gene expression and morphology, and that blockade of IFN-I signaling can rescue some changes in gene expression that correlate with aging in microglia. They then take a candidate approach and show that the transcription factor Mef2C (which is decreased in aged microglia) modulates microglia gene expression, and alters mouse behavior in response to cytokine challenge.

Overall, the manuscript presents several interesting findings linking factors in the aged CSF to changes in microglial state. However, enthusiasm for the manuscript is tempered by a lack of functional data linking microglia to any of the observed phenotypes (synaptic loss, behavior, etc) in aged mice, mice over-expressing IFN-B in the CNS, or mice lacking Mef2C. The story becomes difficult to follow because the authors suggest so many potential pathways by which IFN-I or loss of Mef2C could modulate brain homeostasis (for example, upregulation of complement C4 and B2M, down-regulation of neurotropic factors, uncontrolled production of inflammatory cytokines) without going into any of these potential mechanisms in depth. This manuscript would be much more impactful if the authors focused on one pathway linking changes in the CSF to changes in microglia profile, and then directly connected these microglia-specific changes to a neuronal phenotype underlying behavioral changes in mice using appropriate, cell or cellular process-specific knockouts. This is a major weakness given previous reports demonstrating age-related changes in microglia transcriptional profiles that suggest alteration of microglia homeostasis, and evidence that B2M is an age-related factor associated with impaired neurogenesis and cognitive functions. Moreover, I have a number of concerns relating to approach and lack of rigor that further diminish impact and enthusiasm as summarized below.

Major comments:

- In Figure 1, the authors show that i.c.v. injection of AAV1-IFN-B infects the choroid plexus of the lateral ventricle. However, it is unclear whether the virus also infects other cells in the brain, or the brain endothelium. To address this point, it is necessary to show a representative overview of expression pattern of AAV1 after injection. This is particularly important because the authors use induced IFN-B expression in the choroid plexus to mimic natural upregulation of IFN-B in the choroid plexus with ageing.
- The authors should provide evidence that levels of IFN-B produced by the choroid plexus after AAV infection are similar those produced by the aging choroid plexus. This could be done at least by qPCR. A more rigorous analysis would involve culturing AAV-IFN-B or aged choroid plexus (and corresponding controls) ex vivo and analyzing the supernatants for IFN-B, or analyzing IFN-B levels in the CSF by ELISA. If AAV-induced IFN-B production is drastically higher than natural production with aging, this would call into question the relevance of using AAV expression of IFN-B expression as a model of the aging process.
- The authors use IBA-1 reactivity as a measure of microglia activation in Figure 1. Because the intensity of IBA-1 immunoreactivity changes drastically between control and IFN-B overexpressing

conditions, it is difficult to assess changes in other microglia morphological parameters (e.g. process thickness, length, branching). For example, dim IBA-1 staining in thin processes may not reach a threshold that is visible in these images, leading to underestimation of process branching in the control condition. These analyses should be repeated with an independent stain (e.g. P2RY12, Tmem119) that stains processes brightly if the authors want to include these parameters as readout of microglial phenotype. Additionally, these markers are expressed specifically in microglia, which will increase confidence that the authors are quantifying only microglia and not infiltrating monocytes in their analyses.

- In Figure 1g, it is unclear what region of the brain is analyzed. Is there spatial heterogeneity in microglial response to chronic IFN-B from the choroid plexus? Does this change in morphology reflect changes seen in the aged brain?

- In Figure 2e-f, the authors show synapse loss in mice overexpressing IFN-B in the CNS. This is interesting preliminary evidence. However, more in depth analyses of synapse number and quantification of several pre and postsynaptic markers are needed. Functional data, ideally electrophysiology would strengthen this claim. Moreover, the mechanisms underlying synaptic changes are not addressed..

- In Figure 3, the authors show changes in microglial gene expression after anti-IFNAR treatment. Here, it would be informative to link IFN-I blockade to phenotypic readouts presented in Figure 2 (synapse loss, behavior). If the IFN-B overexpression model accurately recapitulates major processes driving this dysfunction in aging, these aging phenotypes should be at least partially rescued by anti-IFNAR treatment in aging mice. Additionally, further experiments should be performed to determine whether IFN-B signals directly to microglia, or if this is an indirect effect that alters other components of the CNS aging milieu.

- It is unclear how the authors are quantifying Mef2C expression from the images presented in panels A-C. In (A and B) the IBA-1 staining is too dim to identify microglia, and it looks like it is mostly inside of the nucleus. Additionally, Mef2C staining is very difficult to see and controls for antibody specificity were not included. The confocal image in (C) does not provide much clarification, raising the question how the authors were able to quantify these differences. More evidence of baseline microglia expression of Mef2C and changes in aging and after perturbations are needed. Improvements in the resolution of staining and imaging, or alternative methods, are required to quantify MEF2C protein in microglia.

- It is increasingly appreciated that microglia cultured in vitro do not accurately recapitulate the transcriptional and functional profile of cells in the brain environment. The authors emphasize this point by referencing several papers that demonstrate how tissue-derived signals greatly influence microglial state and function. Therefore, it is questionable whether in vitro transcriptional data from microglia after knockdown of Mef2c can provide significant insight into how lack of Mef2c impacts these cells in vivo. This figure should be moved to supplemental, especially because the authors follow up with in vivo data in the subsequent figure.

- In Figure 5d, how does transcription of genes in LPS treated microglia compare to their expression in mock-treated microglia? It is difficult to assess the magnitude of change between MEF2C sufficient and deficient microglia without this baseline control. Additionally, data showing protein expression of these factors would add significantly to the authors' interpretation of the impact of MEF2C on microglia production of neuromodulatory factors relevant for brain function in aging.

- The authors show that PirB is upregulated in Mef2C knockout microglia, and link this to a paper

indicating that PirB is involved in loss of synaptic function in aging. However, the cited paper indicates that PirB is upregulated in neurons in ageing (and explicitly states that PirB is not changed in microglia/macrophages), so it is unclear how this change in microglia is relevant without further data supporting a functional role for PirB in microglia. It would be sufficient for the authors to provide other evidence from the literature if PirB does in fact play microglia-specific role in modulating synapse function.

- The authors make an intriguing point that Mef2C may contribute to immune homeostasis in microglia at steady state. However, the evidence presented (change in surface expression levels of CX3CR1, CD45, and CD11b) is insufficient to support this claim. More appropriate analyses might include intracellular staining for inflammatory cytokines produced by microglia upon knockout of Mef2C, a more thorough analysis of RNA seq data collected for acutely isolated Mef2C versus WT microglia, or RT-qPCR for genes associated with inflammation in Mef2C KO microglia.
- It is unclear why the authors switched immune challenge paradigms in these panels, and how injection of TNF- α is relevant to studying the role of Mef2C in microglia immune homeostasis. The authors should provide justification for this paradigm (e.g. why use i.c.v. injection of TNF- α instead of peripheral injection of Poly I:C or LPS as a model for immune activation).
- The authors have gone through considerable effort to breed mice conditionally lacking Mef2C in microglia/Cx3cr1-expressing cells. However, the analyses presented (RNA-seq, one behavioral assay) indicate a missed opportunity to investigate the biology of Mef2C in microglia as it relates to IFN- β -driven or aged phenotypes presented earlier in the paper. A cellular or molecular readout linking loss of Mef2C in microglia to dysfunction in the brain driven by aging CSF would greatly improve the clarity and continuity of this manuscript.
- The authors should be careful of how they interpret their findings from Figure 6. For example, they state that: "...animals exposed to TNF- α exhibited a reduced preference for social interaction, and this was further decreased in Mef2C-KO animals (Fig. 7g, Supplementary fig. 6k) indicating an unrestrained response...". Here, it is unclear how a change in social preference can indicate an "unrestrained [immune] response." Data indicating exaggerated production of inflammatory cytokines, or other metrics of microglial immune reactivity are required to make this claim. More rationale for choice of behavioral tests are also needed.
- In the discussion, the authors comment on potential implications of IFN-I signaling in developmental disorders such as autism and schizophrenia, suggesting that IFN-I induced by maternal immune activation (MIA) could cause microglial dysfunction that leads to behavioral deficits. While this is an intriguing hypothesis, more evidence should be required before the authors can claim that blockade of IFN-I signaling during early development might "rescue" the MIA phenotype. There is little evidence in the field that IFN-I drives the MIA phenotypes. Additionally, several publications have demonstrated that microglia undergo significant phenotypic and functional changes from early postnatal development to adulthood, which makes it difficult to conclude that microglial responses to a particular stimulus will be conserved in aged mice and developing mice.

Additional comments:

- The term "cognition-impairing" is used throughout the paper to describe genes and proteins. This gives the impression that the main role of these factors is to disrupt brain function, when they are most likely integral to normal brain development and maintenance when expressed at homeostatic levels (e.g. role for B2M and C4b in synaptic pruning). The term should be replaced or removed.

Figure 6

- f) change "CTLR" to "CTRL"
- g) this is referred to as "Figure 7g" in the text, and the corresponding supplemental figure is referred to as "Suppl. 6k" (should be suppl. 7k)

Supplemental Figure 3.

- e-f) missing units on Y axis
- g) How well is this algorithm validated for analyzing microglia morphology? How does this to classic measures of microglia state (CD68/morphology)? Additionally, these representative images are not taken in similar anatomical locations – this should be corrected, and the authors should indicate where these images are located with a schematic of the brain or image inset.

Supplemental Figure 7

- It would be informative to see representative images of Mef2C knockout in vivo. This may be best assessed by FACS. Also, microglia could be sorted and RT-qPCR used to evaluate extent of knockdown/knockout.
- f-g) Change "open filed" to "open field"

Reviewer #3 (Remarks to the Author):

This is a well written paper with well conducted experiments adequately controlled.

1. Microglia defined by CD11b,CD45int had over 600 differentially expressed genes between old and young mice.
 2. Of interest 60% (224 out of 356) of genes,were significantly upregulated in aging mice were induced by IFN-I
 3. Using AAV transfected mice, they showed that "there was a significant overlap between genes with increased expression in microglia isolated from AAV-IFN β infected mice and of microglia from aged mice, and between the genes with a decreased expression in microglia of AAV-IFN β and aged animals (Fig. 1f)."
 4. They "tested whether the continuous production of excessive IFN-I is sufficient to accelerate functional aging of the brain": "Middle-aged mice infected with AAV-IFN β virus failed to identify novel locations in the NLR test (Fig. 2b), committed more errors, and took a significantly longer time to locate the target platform relative to mice infected with the empty virus in the RAWM task (Figs. 2c and 2d)."
 4. IFNAR blockaderesulted in changes in global gene expression that were inversely related to those found in the aging phenotype at all three time-points (Figs. 3b and 3c, Supplementary figs. 5a-c, Supplementary table 3)
 - 5.Reduction of Mef2C expression in cultured microglia led to increased production of pro-inflammatory chemokines
 6. mice exposed to TNF- α exhibited a reduced preference for social interaction, and this was further decreased in Mef2C-KO animals (Fig. 7g, Supplementary fig. 6k) indicating an unrestrained response and worse recovery from an inflammatory stimulus.
- Overall, an impressive set of experiments and results.

Reviewers' comments:

Reviewer #1 (Remarks to the Author):

This revised manuscript has been substantially improved. It flows well and the more succinct presentation of information together with the additional data makes this a very interesting paper.

Reviewer #2 (Remarks to the Author):

This is an interesting study and the revised paper includes convincing new data showing that deletion of IFNAR signaling in CX3CR1+ cells (including microglia) reverses behavioral deficits imposed by IFN-B over-expression in the choroid plexus. However it is still unclear how IFN-B signaling in microglia leads to synapse loss and behavioral deficits or how IFN-B dependent downregulation of MEF2C leaves microglia vulnerable to "pro-inflammatory stimuli in the aged brain." Please see below for remaining concerns and suggestions.

- The authors show that IFN-B over-expression leads to loss of VGLUT2 or synaptophysin area and intensity in the CA1 and CA3 of the hippocampus, and that deletion of IFNAR in CX3CR1+ cells rescues this phenotype. However, synaptic staining intensity is not a valid measure synapse number. For example, in Figure 3 k-l, the authors claim that IFNAR KO in CX3CR1+ cells rescues synapse loss is not supported by this data. Quantification of synapses (ie by co-localization of pre- and post-synaptic elements) or ideally functional data is needed.
- In Figure 4, the addition of the confocal image of IBA-1 and MEF2C staining does not add much to this analysis. The IBA-1 staining is still of poor quality. Please provide higher resolution images which is needed to analyze and quantify MEF2C expression.
- The authors propose that IFN-B dependent down-regulation of MEF2C leaves microglia vulnerable to "pro-inflammatory stimuli that prevail in the aged brain." While this is an interesting hypothesis, the authors do not show data to support this. What are the "pro-inflammatory stimuli" that prevail in the aging brain? Further, the link between IFN-B dependent MEF2C down-regulation to and changes in synapse numbers and learning/memory behaviors are correlative. More evidence to support this link would strengthen the paper. Moreover does TNF-a injection also alter synapse numbers or learning/memory behavior?

Minor comments

- The mouse C4a gene (otherwise known as SLP) is not homologous to human C4a (an isoform of complement component C4 implicated in risk for schizophrenia). The only C4 component that mice have is C4b. Thus, it is confusing how the authors can use an anti-mouse "C4a/b" antibody (Supplemental Figure 9) and claim that they are staining for mouse complement component C4. Has this antibody been tested in mouse C4 KO tissue? Please clarify. Please clarify.

REVIEWERS' COMMENTS:

Reviewer #2 (Remarks to the Author):

This is an interesting paper that will be of broad interest to the field.

The authors have addressed most of my concerns in the revised paper; however I have a few remaining comment related to the synaptic analyses that need to be addressed as outlined below:

As stated in my previous review, more direct evidence for IFNB dependent changes in functional or structural synapses would strengthen the paper and its impact. Synaptophysin or vGlut2 intensity should not be used to quantify synapse loss or synapse numbers as intensity can change for many reasons that have nothing to do with changes in the number of synapses present. The evidence that they present (Fig. 3k,l, Suppl. Fig 7c) for rescue of synapse loss in mice lacking IFNAR in CX3CR1+ cells, and data as preempted are not convincing.

Providing quantitative analyses of structural synapse loss using established methods to measure co-localization of pre and post synaptic markers should be performed. If this is not feasible on existing tissue, then the terminology in text and in Figure 3 (including legend) should be changed to presynaptic puncta or something along those lines, so as not to suggest altered synapse numbers in these studies. Synaptophysin and other presynaptic proteins often appear as puncta in transport packets and vesicles along the axon and are thus not a marker of presynaptic terminals.

-In figure 3k, it looks like synaptic puncta are being quantified in different areas of CA3 in their different groups. The representative images they have chosen have many more cell bodies in the experimental group compared to the control, so the density looks lower just by virtue of all the blank space taken up by cell bodies. Please clarify the method of analyses and imaging.

- Please provide controls or evidence of specificity for mouse C4 immunostaining shown in Supplemental Fig 9h.

Reviewers' comments:

We thank the reviewers for their in-depth constructive comments. We believe that addressing the issues that they raised markedly improved our study. All revised parts of the text were highlighted in yellow.

Reviewer #1 (Remarks to the Author):

This manuscript considers the functional overlap between age, IFN β (in the choroid plexus) and downregulation of a transcript factor, which the authors suggest is a key factor in modulating microglial activation. A good deal of data is presented, some of which is predictable based on the authors previous work, or not especially new. What is new, is the finding that Mef2C can exert a modulatory effect on microglia but this aspect of the work is under-developed and preliminary, and the impact of age and IFN β on its expression, the consequent microglial activation and neuronal function needs further work.

We appreciate the reviewer's comment. We would like to emphasize that beyond the finding of Mef2C role in microglia, which we now support with additional data (Fig.3-6), the manuscript describes additional novel findings: 1. Establishes previously unknown link between IFN-I present in aged brain microenvironment and aging microglia phenotype and 2. Mechanistically links IFN-I in the aged brain, microglial response to it, and its effect on brain function. Specifically, as we describe in detail below, using microglia-specific genetic deletion of IFNAR, we now show that the effect of IFN-I on the brain is mediated by microglia (Fig.3).

Specific major issues:

The abstract

1. makes reference to a microglial "homeostatic phenotype" but this is not properly defined anywhere in the text;

We agree with this comment, in the revised manuscript we clarified our statements and we now better explain this term in the introduction.

2. states that the 'aged microglial phenotype is imposed by the cerebrospinal fluid microenvironment' but this is not shown;

We now use more precise terms to describe this phenomenon, stressing the role of IFN-I.

3. concludes that "detrimental aging-related activity of microglia to be largely imposed by the aged brain microenvironment, and identified Mef2C as a key transcription factor regulating microglial homeostatic function under immune challenge", but only the second part of this is actually demonstrated.

The abstract is now rewritten, and all comments have been addressed.

Lines 107-109: It is stated that IFN β confined to brain - it would be better to state that it is confined to the choroid plexus since there was no change in the hippocampus and, with respect to the persistent response at 7 months, this was evaluated only in CP so being specific in the first place is appropriate.

We agree with these remarks; we now clarified this point in the text.

The implication is that IFN β stimulates microglial activation in the hippocampus but there is no evidence of increased IFN β in the brain tissue so the authors need to spell out how changes can be attributed to IFN β .

We observed a non-significant, increasing trend of IFN-I response in hippocampi of mice with AAV-dependent overexpression of IFN-I (Supplementary fig. 4e), yet analysis of microglia showed statistically significant induction of IFN-I response (Fig. 2c,d, Supplementary fig. 4i, Supplementary fig. 7a). These results suggest that the response to IFN-I expressed by the choroid plexus in the brain might be mediated by microglia.

Further, in the revised manuscript, using AAV-mediated IFN β overexpression in mice lacking IFNAR specifically on microglia (mic-IFNAR-KO; IFNAR fl/fl::CX3CR1-ERT2-Cre) we now show that IFN β affects microglial phenotype directly. For example, overexpression of IFN-I in mic-IFNAR-KO did not change microglial expression levels of IFN-I-dependent genes (Supplementary fig. 7a), including B2m, C4b (Supplementary fig. 9j-m) and Mef2C (Fig.4d, e). In addition, mice lacking IFNAR on their microglia were protected from IFN-I-induced decline of hippocampal-dependent brain function and synaptic loss in CA1 and CA3 (Fig 3.i-i, Supplementary fig.7c), what further suggests a critical role of microglia in transmitting the effect of IFN-I on hippocampal functions.

With respect to Fig 1g, labelling indicates changes in hippocampus; do the data presented in Fig 1h related to the hippocampus (the whole hippocampus or a sub-field)? This should be stated.

The sequencing data were obtained from the whole brain microglia, which is now emphasized in the main text and in the Online Methods section. In Fig. 2a and Supplementary figs 5a and 5f we now present the subfields of hippocampus and cortex used for Iba1 quantification. In addition, we now included zoom out pictures (Fig 2e, Supplementary fig. 5g).

Within the brain, we observed regional diversity in terms of IFN-I-induced alterations in microglial morphology: hippocampal microglia, located closer to the IFN-I expression site had clearly changed, aging-like morphology, while cortical microglia retained their homeostatic morphology.

In this context, VgluT2 was assessed in area CA3; why was this area selected for analysis and what changes, if any were found in other sub-fields?

In the brain function analysis we focused on the hippocampus, activity of which is critically affected by aging. Therefore, we tested mice using hippocampal-dependent cognitive tests and quantifies synapses in subfields of hippocampal slices. Specifically, in the revised manuscript we added quantification of VgluT2 in CA1, and Synapthophysin in CA1 and CA3 (Fig. 3e-h) and quantified the

same markers (and fields) in mic-IFNAR-CTRL and mic-IFNAR-KO mice infected with AAV-IFN β or AAV-CTRL(Fig. 3k-l, Supplementary fig. 7c).

Loss of Vglut2 in CA3 and CA1 was previously associated with aging-related cognitive loss (Morrison, Nature Reviews Neuroscience, 2012), similarly to synaptophysin, the loss of which correlated with aging (revised in Morrison and Baxter, Nat.Rev.Neurosci. 2012) and impaired performance in a hippocampus-dependent cognitive task (Calhoun, Neurobiol Aging 1998). We now included this explanation in the main text and added a micrograph illustrating CA1 and CA3 areas, which we quantified (Fig 3a).

No details are given about what area of the brain was used to prepare microglia (Fig 2g and 2k), nor what area was assessed for β 2M or C4a/b immunoreactivity (Fig 2i and 2m).

As now indicated in the Online Methods section and throughout the manuscript, microglia for RNA-Seq were isolated from the whole brain. Immunoreactivity was evaluated in the hippocampus; representative images of areas from which the β 2M or C4a/b immunoreactivity was quantified are shown in Supplementary fig. 9c and 9h.

While the data indicate anti-IFNAR antibody treatment reduced age-related gene expression, it is odd that neither behavior nor microglial activation were evaluated.

Microglia activation status and behavior in anti-IFNAR antibody- treated old mice were tested in our previous work (Baruch et al, Science, 2014), which created the basis for the current manuscript. In addition, we now tested cognitive ability of aged mice genetically lacking IFN-I signaling (Supplementary fig. 8a), however, the goal of the present study was to test whether chronic expression of type I interferon in the brain's choroid plexus is sufficient to cause aging-like changes, and if so, whether its pathway involves a direct effect on brain microglia, and what are the microglial factors and/or functions modulated by IFN-I in aged brain microenvironment.

Mef2C becomes the focus of attention from Figure 4 and this shift is somewhat odd given that anti-IFNAR antibody did not significantly rescue the age-related change (although decreases with age and IFN β overexpression were observed).

We appreciate the referee's comment and in the revised manuscript we now explain the rationale that prompted us to test the role of Mef2C in microglia In short, we aimed at finding a transcription factor whose levels would change in the presence of IFN- β and in aging. Mef2C fulfilled this requirement, as its expression levels were negatively correlated with expression of IFN-I-dependent genes in IFN-I sensitive microglia (mic-IFNAR-CTRL mice) but not in microglia of mic-IFNAR-KO mice (Fig. 4c-e). Importantly, Mef2C was previously suggested to regulate microglial biology (Lavin, Cell, 2014), and human GWAS linked Mef2c mutations with aging-associated late-onset Alzheimer's disease (Lambert, 2013), however the function of Mef2C in microglia has never been addressed experimentally. The combination of these features made it an attractive candidate for further studies.

With respect to Fig 4, what brain area was used to evaluate Mef2C mRNA?

We used whole brain microglia to obtain RNA for Mef2C expression evaluation, and this is now clarified in the Online Methods section and in the text.

The impact of downregulating Mef2C was assessed in vitro and in vivo but the readouts were different for the 2 experiments and it would certainly strengthen the argument for a key modulatory role for Mef2C if at least some of the same readouts were assessed. It is not clear why spatial learning was not assessed in Mef2C KO mice.

We completely agree with the reviewer. In the revised manuscript, we removed most of the in vitro experiments and substantiated our initial in vivo findings of Mef2C function with additional data (Figs 6, Supplementary fig. 11).

In the revised manuscript, we tested Mef2C-KO and CTRL mice, using a battery of behavioral tests for assessment of hippocampal-dependent spatial learning, working and long-term memory, social preference and anxiety, and we did not observe any behavioral deficit that could be attributed to microglial Mef2C deficiency (Supplementary fig. 11b-e). In contrast, upon challenge with an immune stimulus Mef2C-KO mice showed behavioral deficits and exaggerated inflammatory response in microglia (Fig.6 a-g).

Fig 6a,b: It is not acceptable to present SEM values or to undertake statistical analysis when data are derived from only 2 replicates.

We thank for this comment and apologize for the oversight. We repeated the sequencing of microglia in Mef2C-KO mice (and CTRL mice) using larger cohorts (n=5 per group; Supplementary table 5) and the results supported our initial findings; namely, we did not observe any strong transcriptional signature in Mef2C-KO mice under steady state conditions, what additionally encouraged us to test the function of Mef2C upon inflammatory challenge (Fig.6).

Reviewer #2 (Remarks to the Author): Deczkowska, Matcovitch-Natan et al. present evidence for a CSF-microglia axis that potentially modulates microglial homeostasis during aging. Through a series of over-expression and antibody blocking experiments, they demonstrate that elevated IFN- β production in the brain is sufficient to drive changes in microglia gene expression and morphology, and that blockade of IFN- α signaling can rescue some changes in gene expression that correlate with aging in microglia. They then take a candidate approach and show that the transcription factor Mef2C (which is decreased in aged microglia) modulates microglia gene expression, and alters mouse behavior in response to cytokine challenge.

Overall, the manuscript presents several interesting findings linking factors in the aged CSF to changes in microglial state. However, enthusiasm for the manuscript is tempered by a lack of functional data linking microglia to any of the observed phenotypes (synaptic loss, behavior, etc) in aged mice, mice over-expressing IFN- β in the CNS, or mice lacking Mef2C. The story becomes difficult to follow because the authors suggest so many potential pathways by which IFN- α or loss of Mef2C could modulate brain homeostasis (for example, upregulation of complement C4 and B2M, down-regulation of neurotropic factors, uncontrolled production of inflammatory cytokines) without going into any of these potential mechanisms in depth. This manuscript would be much more impactful if the authors focused on one pathway linking changes in the CSF to changes in microglia profile, and then directly connected these microglia-specific changes to a neuronal phenotype underlying behavioral changes in mice using appropriate, cell or cellular process-specific knockouts. This is a major weakness given previous reports demonstrating age-related changes in microglia transcriptional profiles that suggest alteration of microglia homeostasis, and evidence that B2M is an age-related factor associated with impaired neurogenesis and cognitive functions. Moreover, I have a number of concerns relating to approach and lack of rigor that further diminish impact and enthusiasm as summarized below.

We thank this reviewer for these constructive comments. The manuscript has been re-organized and, we believe, it now presents the data in a more accessible, easier-to-follow way. In the manuscript, we do mention and show evidence for activation of several pathways upon IFN- α presence in the aged brain microenvironment (now mostly in the 'Supplementary' section), but we clearly focused on one specific factor, Mef2C, and we now performed additional in vivo experiments that further substantiate our initial observations regarding its role in microglia (Fig 4-6). In addition, to address a major concern of this reviewer, we now combined a mouse model of microglia-specific ablation of IFN- α receptor (IFNAR $fl/fl::Cx3CR1-ERT2$ -Cre mice) with IFN overexpression to demonstrate the impact of microglia-specific IFN- α response on the brain (Fig 3i-k, Fig.4d,e, Supplementary figs.7, 9j-m).

Major comments:

- In Figure 1, the authors show that i.c.v. injection of AAV1-IFN- β infects the choroid plexus of the lateral ventricle. However, it is unclear whether the virus also infects other cells in the brain, or the brain endothelium. To address this point, it is necessary to show a representative overview of expression pattern of AAV1 after injection. This is particularly important because the authors use induced IFN- β expression in the choroid plexus to mimic natural upregulation of IFN- β in the choroid plexus with ageing.

To quantitatively analyze infection rate in the choroid plexi and the brain parenchyma (especially hippocampus, function of which was analyzed throughout the study) we separated the choroid plexi of third, lateral (injected and non-injected) and fourth ventricles, and the hippocampus of the injected ventricle, and measured levels of virus-derived GFP using qPCR (Supplementary fig. 4b,c). We clearly observed high levels of GFP mRNA in all choroid plexi of AAV-CTRL and AAV-IFN-I-injected mice, whereas no change was observed in the hippocampus of the injected ventricle, where unintended virus overexpression due to an imprecise injection would be most probable. Overall these data suggest that the IFN-I expression was restricted to the choroid plexus, and that the changes observed in microglia and hippocampus in AAV-IFN β -infected mice were caused by choroid plexus-derived IFN-I.

- The authors should provide evidence that levels of IFN-B produced by the choroid plexus after AAV infection are similar those produced by the aging choroid plexus. This could be done at least by qPCR. A more rigorous analysis would involve culturing AAV-IFN-B or aged choroid plexus (and corresponding controls) ex vivo and analyzing the supernatants for IFN-B, or analyzing IFN-B levels in the CSF by ELISA. If AAV-induced IFN-B production is drastically higher than natural production with aging, this would call into question the relevance of using AAV expression of IFN-B expression as a model of the aging process.

We thank the reviewer for this comment. In the revised manuscript we present the data suggesting that even though the local levels of IFN-I response in the CP seems very high in the IFN-I-overexpressing mice, the changes induced in microglia, which are the main focus of our study, are comparable to those observed in aging. Specifically, we compared the expression levels of IFN-I dependent genes in model of aging and AAV-mediated overexpression of IFN-I. Whereas in aging the choroid plexus showed 4-6 fold induction of Irf7 and 12-15 fold induction of Ifit1 (Baruch et al, Science 2014; Fig 1b), the fold induction changes observed in AAV-infected mice CPs were much higher (approximately 480 fold induction of Irf7 and 960 fold induction of Ifit1; Supplementary fig. 4d). However, aging microglia showed 2.6x induction of Ifit1, 2x induction of Isg15, and 3.3x induction of Ifi204, and microglia of IFN-I-overexpressing mice displayed 3x induction of ifit1, 3,5x induction of Isg15, and 3.3x induction of Ifi204 (Supplementary fig .4i; based on RNA-Seq in Supplementary tables 1,3,4, all data-points are shown as relative to their controls; young or AAV-CTRL mice, respectively).

- The authors use IBA-1 reactivity as a measure of microglia activation in Figure 1. Because the intensity of IBA-1 immunoreactivity changes drastically between control and IFN-B overexpressing conditions, it is difficult to assess changes in other microglia morphological parameters (e.g. process thickness, length, branching). For example, dim IBA-1 staining in thin processes may not reach a threshold that is visible in these images, leading to underestimation of process branching in the control condition. These analyses should be repeated with an independent stain (e.g. P2RY12, Tmem119) that stains processes brightly if the authors want to include these parameters as readout of microglial phenotype. Additionally, these markers are expressed specifically in microglia, which will increase confidence that the authors are quantifying only microglia and not infiltrating monocytes in their analyses.

To address this comment, in the revised manuscript we replaced the Iba-1 staining micrographs, and we are now able to better show the observed changes in microglial morphology (Fig 2e, Supplementary fig. 5).

It is true that Iba-1 can also stain macrophages, which can be clearly seen as brighter, but these are found only in the vicinity of the choroid plexus and meninges under physiological conditions, and these areas were not analyzed. To address the reviewer's comments, we inserted additional representative pictures showing how the algorithm for microglial morphology analysis recognized that young / AAV-Ctrl microglia exhibit weaker Iba1-staining, but longer processes relative to aged or AAV-IFN β microglia, suggesting that even weak Iba-1 staining was correctly recognized and quantified.

- In Figure 1g, it is unclear what region of the brain is analyzed. Is there spatial heterogeneity in microglial response to chronic IFN-B from the choroid plexus? Does this change in morphology reflect changes seen in the aged brain?

Analysis of Iba1 in the subfields of cortex and hippocampus indeed revealed spatial diversity. Microglia in the hippocampus, a site largely affected by aging which is located in physical proximity to the brain ventricles, filled with CSF and harboring the source of IFN-I, the choroid plexi, showed altered Iba1+ cell staining intensity and aging-like morphology (Fig.2e-f). In contrast cortical microglia were largely unaffected (Supplementary fig. 5f-h). In addition, morphological changes observed in hippocampal microglia of the AAV-IFN-b-overexpressing mice (compared to AAV-CTRL) were also observed in aged mice brains (compared to young, non-manipulated mice) (Supplementary fig. 5a-e).

- In Figure 2e-f, the authors show synapse loss in mice overexpressing IFN-B in the CNS. This is interesting preliminary evidence. However, more in depth analyses of synapse number and quantification of several pre and postsynaptic markers are needed. Functional data, ideally electrophysiology would strengthen this claim. Moreover, the mechanisms underlying synaptic changes are not addressed.

In addition to changes in VgluT2 puncta in CA3, we now show and quantify VgluT2 puncta in CA1, and show a similar set of data for Synaptophysin (Fig.3 e-h). To strengthen our results, we show that mice exhibiting loss of synaptic connections in regions of the hippocampus related to memory, show worse performance in hippocampal memory-based tasks including the Radial Arm Water Maze, and Novel Location Recognition (Fig 3b,c). Importantly, mice lacking IFNAR on microglia were protected from the negative impact of IFN-I on hippocampal function, in terms of performance in spatial memory task (Fig. 3j) and number of synaptic terminals (Fig. 3k,l; Supplementary fig. 7c).

We now discuss that synaptic loss can occur due to IFN- β -induced elevation of B2m and C4b expression; both MHC-I and complement factors were implicated in excessive synapse clearance. Moreover, virus-induced synaptic loss, which induces an IFN-I response, was shown to be mediated by complement pathway activation and microglia (Vasek, Nature, 2016). Further, IFN-I-induced reduction in microglial Mef2C may promote stronger microglial activation (microglial "priming"),

and thereby contribute to pro-inflammatory environment, previously linked to excessive synaptic clearance and cognitive loss (Reviewed in: Simen et al., Ther Adv Chronic Dis., 2011).

- In Figure 3, the authors show changes in microglial gene expression after anti-IFNAR treatment. Here, it would be informative to link IFN-I blockade to phenotypic readouts presented in Figure 2 (synapse loss, behavior). If the IFN-B overexpression model accurately recapitulates major processes driving this dysfunction in aging, these aging phenotypes should be at least partially rescued by anti-IFNAR treatment in aging mice. Additionally, further experiments should be performed to determine whether IFN-B signals directly to microglia, or if this is an indirect effect that alters other components of the CNS aging milieu.

We thank the reviewer for this comment. While we observe some analogies between the IFN-I overexpression in young mice and IFN-I blockade in aged mice, these models are different, and their usage served different purposes. The idea here was to test if the mere upregulation of IFN-I in the CP is sufficient to induce IFN-I-dependent genes in the microglia, and impose aging-like effects on the brain function and animal behavior.

The effects of IFN-I induction and blockade are not completely symmetric; for example, whereas in young mice the induction of IFN-I is strong and chronic, blockade of IFNAR under our experimental conditions is transient and likely not fully efficient. However, the fact that we observed clear gene expression signatures in the IFNAR blockade model suggests that soluble IFN- β present in the aged brain milieu shapes aged microglial phenotype, and this finding created the basis for studying the effects of this cytokine on microglia in a cleaner model of IFN-I overexpression in young animals. In our previous study, we showed that transient blockade of IFN-I signaling using i.c.v. injection of anti-IFNAR neutralizing antibody reversed cognitive deficits in aged mice. In these mice, we observed an increase in hippocampal neurogenesis levels and reduced micro- and astrogliosis (suggesting reduced neuroinflammatory state) (Baruch, et al. 2014). In the revised manuscript, to better illustrate this issue, we present the data in a different sequence.

- It is unclear how the authors are quantifying Mef2C expression from the images presented in panels A-C. In (A and B) the IBA-1 staining is too dim to identify microglia, and it looks like it is mostly inside of the nucleus. Additionally, Mef2C staining is very difficult to see and controls for antibody specificity were not included. The confocal image in (C) does not provide much clarification, raising the question how the authors were able to quantify these differences. More evidence of baseline microglia expression of Mef2C and changes in aging and after perturbations are needed. Improvements in the resolution of staining and imaging, or alternative methods, are required to quantify MEF2C protein in microglia.

We now improved our Mef2C staining protocol and observed that nearly all microglia were stained for Mef2C in a young/aged and AAV-IFN β /CTRL mice, but the Mef2C staining in microglia in aged and AAV-IFN- β -overexpressing animals was weaker, as shown now in Figs. 4g-h. In light of these results, we removed the IHC-based Mef2C/Iba1 quantification in a young/aged and AAV-IFN β /CTRL mice from the manuscript. Using the same staining protocol, we observed marked loss of Mef2C staining from microglia of Mef2C-KO mice; only approximately 15% of Iba1+ cells showed Mef2C staining (compared to 97,5% of Iba1+microglia in the CTRL mice; Figure 5a,b).

IN the revised manuscript, we also include a new confocal image, better showing nuclear localization of Mef2C (Fig.4f) and additional controls for Mef2C staining (Secondary (Cy3) only and w/o anti-Mef2C, w/o Cy3) to show specificity of the method (Supplementary fig.10).

- It is increasingly appreciated that microglia cultured in vitro do not accurately recapitulate the transcriptional and functional profile of cells in the brain environment. The authors emphasize this point by referencing several papers that demonstrate how tissue-derived signals greatly influence microglial state and function. Therefore, it is questionable whether in vitro transcriptional data from microglia after knockdown of Mef2c can provide significant insight into how lack of Mef2c impacts these cells in vivo. This figure should be moved to supplemental, especially because the authors follow up with in vivo data in the subsequent figure.

We completely agree with the reviewer. In the revised manuscript we acquired more relevant data using our in vivo Mef2C-KO model, and therefore decided to remove most of the in vitro findings from the manuscript. We also repeated RNA-Seq of Mef2C-KO microglia using more animals (n=5; Supplementary table 5) and tested Mef2C-KO and CTRL mice, using a battery of behavioral tests for assessment of hippocampal-dependent spatial learning and memory, working and long-term memory, social preference and anxiety (Supplementary fig. 12), and we did not observe any behavioral deficit, nor microglial signature, that could be attributed to microglial Mef2C deficiency under physiological conditions. Only upon challenge with an immune stimulus the Mef2C-KO mice showed behavioral deficits and exaggerated inflammatory response in microglia (Fig.6). This aspect of microglial phenotype was observed both in vivo (Fig.6 a-g) and in vitro (Fig.6h).

- In Figure 5d, how does transcription of genes in LPS treated microglia compare to their expression in mock-treated microglia? It is difficult to assess the magnitude of change between MEF2C sufficient and deficient microglia without this baseline control. Additionally, data showing protein expression of these factors would add significantly to the authors' interpretation of the impact of MEF2C on microglia production of neuromodulatory factors relevant for brain function in aging.

To address to this comment, we now show gene expression levels from LSP-treated and untreated wells separately (Fig.6h). The basal levels of expression of the factors measured are very low, and not different between the siControl and siMef2C, suggesting that loss of Mef2C does not induce a pro-inflammatory state, but sensitizes microglia to more robust responses to immune stimuli. An analogous phenotype was observed in vivo (Fig6a-g).

- The authors show that PirB is upregulated in Mef2C knockout microglia, and link this to a paper indicating that PirB is involved in loss of synaptic function in aging. However, the cited paper indicates that PirB is upregulated in neurons in ageing (and explicitly states that PirB is not changed in microglia/macrophages), so it is unclear how this change in microglia is relevant without further data supporting a functional role for PirB in microglia. It would be sufficient for the authors to provide other evidence from the literature if PirB does in fact play microglia-specific role in modulating synapse function.

Due to space limitations and to tighten the focus of the manuscript we now removed PirB.

- The authors make an intriguing point that Mef2C may contribute to immune homeostasis in microglia at steady state. However, the evidence presented (change in surface expression levels of CX3CR1, CD45, and CD11b) is insufficient to support this claim. More appropriate analyses might include intracellular staining for inflammatory cytokines produced by microglia upon knockout of Mef2C, a more thorough analysis of RNA seq data collected for acutely isolated Mef2C versus WT microglia, or RT-qPCR for genes associated with inflammation in Mef2C KO microglia.

We agree with this reviewer that the use of the term ‘microglial homeostasis’ or ‘immune homeostasis’ might have been misunderstood. Our goal was to emphasize the role of Mef2C in restoration of immune homeostasis in the brain following immune challenge. We now support this view with additional data (Fig.6) and use more precise terminology to describe this phenomenon in the text.

- It is unclear why the authors switched immune challenge paradigms in these panels, and how injection of TNF- α is relevant to studying the role of Mef2C in microglia immune homeostasis. The authors should provide justification for this paradigm (e.g. why use i.c.v. injection of TNF- α instead of peripheral injection of Poly I:C or LPS as a model for immune activation).

We agree that our choice of i.c.v. TNF- α administration and social behavior evaluation was not sufficiently explained; we now better describe the rationale behind our experimental approaches and add more data substantiating our conclusion regarding this part.

Because our inducible Cx3cr1-ERT2-Cre model (Goldman, Nat.neurosc. 2013) leads to stable knock out of the floxed gene not only in microglia, but also in other resident macrophages of embryonic origin (Cronk, Immunity, 2015), we specifically aimed at an i.c.v.-based approach, to target the brain and avoid systemic effects. The paradigm that we used was previously well-described in terms of time-dependent resolution (Palin, J.Neuroimmune, 2007); after 24h, wild type mice showed almost full recovery in a social preference test, but Mef2C-KO mice exhibited markedly reduced social behavior. The direct impact of Mef2C deficiency on microglia ability to cope with inflammation was also assessed in vitro using LPS, a classic, strong pro-inflammatory factor widely used in in vitro studies. This is now better described in the manuscript.

- The authors have gone through considerable effort to breed mice conditionally lacking Mef2C in microglia/Cx3cr1-expressing cells. However, the analyses presented (RNA-seq, one behavioral assay) indicate a missed opportunity to investigate the biology of Mef2C in microglia as it relates to IFN- β -driven or aged phenotypes presented earlier in the paper. A cellular or molecular readout linking loss of Mef2C in microglia to dysfunction in the brain driven by aging CSF would greatly improve the clarity and continuity of this manuscript.

We thank the reviewer for this extremely constructive comment. In response, we now include more data on Mef2C-KO mice. Basically, as shown in Supplementary fig.12, Mef2C-KO mice did not show any behavioral phenotype, nor transcriptional signature in their microglia, as assessed by RAWM, T maze, Y maze, social preference and open field arena tests, and RNA-Seq of acutely isolated microglia. Mef2C-KO microglia appeared slightly (but consistently) more activated when analyzed by flow cytometry (Fig 5b-d) and RNA-Seq did not reveal any strong transcriptional signature of

Mef2C-loss under physiological conditions. These results led us to formulate a hypothesis that deficiency in Mef2C may be critical only in aging or under pathological conditions, often associated with an immune deviation, where this factor may be involved in curbing the inflammatory response, a function that is vitally important in an aging brain, containing various endogenous immunologically-relevant ‘danger signals’.

To test this hypothesis, we used the immune challenge paradigm, and we observed markedly increased microglial activation and worsened behavioral performance in Mef2C-KO mice when compared to CTRL (Fig.6), suggesting that in presence of immune stimuli (e.g. in aging), microglia will respond more robustly if Mef2C is reduced, thereby linking levels of microglial Mef2C, microglial activation, aging brain milieu, and effect on brain function.

- The authors should be careful of how they interpret their findings from Figure 6. For example, they state that: “...animals exposed to TNF- α exhibited a reduced preference for social interaction, and this was further decreased in Mef2C-KO animals (Fig. 7g, Supplementary fig. 6k) indicating an unrestrained response...”. Here, it is unclear how a change in social preference can indicate an “unrestrained [immune] response.” Data indicating exaggerated production of inflammatory cytokines, or other metrics of microglial immune reactivity are required to make this claim. More rationale for choice of behavioral tests are also needed.

We agree with the reviewer, and in the revised manuscript we added more data to substantiate our conclusions. We now investigate the microglia of TNF- α -exposed Mef2C-KO and CTRL mice using flow cytometry and immunohistochemistry; data presented in Fig. 6d-h in the revised manuscript corroborate the claim that Mef2C restrains excessive microglial responses to immune challenge.

As mentioned above, we also better explain in the text the rationale for our choice of the immune challenge.

- In the discussion, the authors comment on potential implications of IFN-I signaling in developmental disorders such as autism and schizophrenia, suggesting that IFN-I induced by maternal immune activation (MIA) could cause microglial dysfunction that leads to behavioral deficits. While this is an intriguing hypothesis, more evidence should be required before the authors can claim that blockade of IFN-I signaling during early development might “rescue” the MIA phenotype. There is little evidence in the field that IFN-I drives the MIA phenotypes. Additionally, several publications have demonstrated that microglia undergo significant phenotypic and functional changes from early postnatal development to adulthood, which makes it difficult to conclude that microglial responses to a particular stimulus will be conserved in aged mice and developing mice.

We fully agree with the reviewer’s comment. To maintain the clarity of the message and due to space limitations, we now removed this part of the discussion in the revised manuscript.

Additional comments:

- The term “cognition-impairing” is used throughout the paper to describe genes and proteins. This gives the impression that the main role of these factors is to disrupt brain function, when they are

most likely integral to normal brain development and maintenance when expressed at homeostatic levels (e.g. role for B2M and C4b in synaptic pruning). The term should be replaced or removed.

We agree with the reviewer's comment. The term 'cognition-impairing' was removed and more accurate terms are now used.

Figure 6

- f) change "CTLR" to "CTRL"
- g) this is referred to as "Figure 7g" in the text, and the corresponding supplemental figure is referred to as "Suppl. 6k" (should be suppl. 7k)

Supplemental Figure 3

- e-f) missing units on Y axis

These errors are all corrected in the revised figures.

- g) How well is this algorithm validated for analyzing microglia morphology? How does this to classic measures of microglia state (CD68/morphology)? Additionally, these representative images are not taken in similar anatomical locations – this should be corrected, and the authors should indicate where these images are located with a schematic of the brain or image inset. ***The NeuroMath program and this specific algorithm has not been used before for analyzing microglial morphology; however, we clearly observed that the numbers obtained accurately reflected the morphology seen in the images (Fig.e 2f and and Supplementary Fig.5a-e).***

We now include representative images from similar locations and mark the line of the dentate gyrus for better clarity, as suggested by the reviewer. In addition, we would like to stress that we evaluated 5 coronal hippocampal slides per mouse; selection of slices encompassed different depths spanning 720nm, and we obtained 10-50 analyzed cells per slide. The incorrectly recognized cell shapes were manually excluded. These measurements (50-250 data points per mouse) were then averaged and compared (n=5 mice per group). We believe that this gives us an accurate representation of the microglial morphology throughout the hippocampus.

Supplemental Figure 7

- It would be informative to see representative images of Mef2C knockout in vivo. This may be best assessed by FACS. Also, microglia could be sorted and RT-qPCR used to evaluate extent of knockdown/knockout.

We thank the reviewer for this suggestion. In the revised manuscript we show data from Mef2C-KO mice (and controls) in which recombination was induced using an optimized tamoxifen regiment (five doses of 4mg tamoxifen every other day) (Fig 5a,b). In addition, as mentioned above, our optimized Mef2C staining protocol revealed that almost all microglia were stained with Mef2C (97,5% of Iba1+ cells) (Fig 5b). Thanks to these improvements we could unambiguously

discriminate between Mef2C+ and Mef2C- microglia in IHC staining, as can be seen in revised Fig. 5a. Quantification revealed ~85% loss of Mef2C in Mef2C-KO microglia (Fig 5b).

- f-g) Change “open filed” to “open field”

This error is now corrected.

Reviewer #3 (Remarks to the Author):

This is a well written paper with well conducted experiments adequately controlled.

1. Microglia defined by CD11b,CD45^{int} had over 600 differentially expressed genes between old and young mice.

2. Of interest 60% (224 out of 356) of genes,were significantly upregulated in aging mice were induced by IFN-I

3. Using AAV transfected mice, they showed that "there was a significant overlap between genes with increased expression in microglia isolated from AAV-IFN β infected mice and of microglia from aged mice, and between the genes with a decreased expression in microglia of AAV-IFN β and aged animals (Fig. 1f)."

4. They "tested whether the continuous production of excessive IFN-I is sufficient to accelerate functional aging of the brain": "Middle-aged mice infected with AAV-IFN β virus failed to identify novel locations in the NLR test (Fig. 2b), committed more errors, and took a significantly longer time to locate the target platform relative to mice infected with the empty virus in the RAWM task (Figs. 2c and 2d)."

4. IFNAR blockaderesulted in changes in global gene expression that were inversely related to those found in the aging phenotype at all three time-points (Figs. 3b and 3c, Supplementary figs. 5a-c, Supplementary table 3)

5.Reduction of Mef2C expression in cultured microglia led to increased production of pro-inflammatory chemokines

6. mice exposed to TNF- α exhibited a reduced preference for social interaction, and this was further decreased in Mef2C-KO animals (Fig. 7g, Supplementary fig. 6k) indicating an unrestrained response and worse recovery from an inflammatory stimulus.

Overall, an impressive set of experiments and results.

We thank this reviewer for these comments.

Dear Huang, PhD

Thank you for your help and support of our manuscript "MEF2C restrains microglial responses to inflammatory stimuli and is reduced in the IFN-I milieu of the aging brain". We are pleased to learn that you are interested in the possibility of publishing our study in Nature Communications. As suggested, below we address the points left by reviewer #2.

Reviewer #2 (Remarks to the Author):

1. This is an interesting study and the revised paper includes convincing new data showing that deletion of IFNAR signaling in CX3CR1+ cells (including microglia) reverses behavioral deficits imposed by IFN-B over-expression in the choroid plexus. However it is still unclear how IFN-B signaling in microglia leads to synapse loss and behavioral deficits or how IFN-B dependent downregulation of MEF2C leaves microglia vulnerable to "pro-inflammatory stimuli in the aged brain." Please see below for remaining concerns and suggestions.

We thank this reviewer for these comments. We now explicitly emphasize in the text, that IFN-I signaling in microglia may affect brain function through several mechanisms, including increased expression of B2m and complement factor C4, genes previously linked to cognitive impairment and loss of synapses (Smith, NatMed, 2014; Sekar, NatMed, 2016; Shi, JNeurosci, 2015) (p.11). Regarding the linkage between Mef2c, Type I interferon, and cognitive loss, we have now more clearly explained that elevation of Type I interferon with aging is linked to reduction of microglial Mef2c, and we further showed that loss of Mef2C expression in microglia led to increased susceptibility to pro-inflammatory challenge, a phenomenon observed also in brain aging and known as "microglia priming" (Norden, Goodbout, Neuropathol Appl Neurobiol. 2013, Simen, et al Therapeutic advances in chronic disease, 2011). This may lead to exacerbation of pro-inflammatory conditions in the aging brain and thereby promote cognitive loss (Guinta, J.Neuroinflammation, 2008). We corrected the text to clarify this issue (p.11-12).

2. The authors show that IFN-B over-expression leads to loss of VGLUT2 or synaptophysin area and intensity in the CA1 and CA3 of the hippocampus, and that deletion of IFNAR in CX3CR1+ cells rescues this phenotype. However, synaptic staining intensity is not a valid measure synapse number. For example, in Figure 3 k-l, the authors claim that IFNAR KO in CX3CR1+ cells rescues synapse loss is not supported by this data. Quantification of synapses (ie by co-localization of pre- and post-synaptic elements) or ideally functional data is needed.

We thank the reviewer for pointing this out.

Indeed, we did not quantify synapses (co-localized post-and presynaptic terminals) but only presynaptic puncta of Vglut2 and Synaptophysin staining intensity. In the revised version of the manuscript we corrected the sentence (p. 7) to be more precise:

"Synaptophysin immunostaining in CA1 and CA3 revealed that mic-IFNAR-KO mice were protected from IFN- β -induced loss **of presynaptic terminals** (Fig. 3k, l, **Supplementary fig 7c**)."

- In Figure 4, the addition of the confocal image of IBA-1 and MEF2C staining does not add much to this analysis. The IBA-1 staining is still of poor quality. Please provide higher resolution images which is needed to analyze and quantify MEF2C expression.

As neurons produce high levels of Mef2C (as can be seen in our Mef2C staining) and microglia are phagocytic, we wanted to confirm that the mRNA of Mef2C, which we identified in microglia, gives rise to a functional transcription factor in this cell type and is not present in our sequencing data solely due to engulfment of neuronal cell debris containing Mef2C. To this end, we performed confocal microscopy, confirming nuclear localization of the Mef2C in microglia.

Following the initial suggestions of this reviewer, we replaced the Iba 1 images in the paper, to better show microglial morphology. We are convinced that the reviewer's description of "poor quality" images was based on the low resolution pictures in the merged pdf file of the manuscript, and not the full resolution figures (files named Figures 1-6 and Supplementary 1-13), in which Iba-1 marked microglial morphology and Mef2C staining can be appreciated.

3. The authors propose that IFN-B dependent down-regulation of MEF2C leaves microglia vulnerable to "pro-inflammatory stimuli that prevail in the aged brain." While this is an interesting hypothesis, the authors do not show data to support this. What are the "pro-inflammatory stimuli" that prevail in the aging brain?

In fig.6 we specifically show that Mef2C-KO microglia are more susceptible to pro-inflammatory stimulus, TNF- α . We chose this immune stimulus based on the previous reports, showing elevated expression of TNF- α in aged brain tissue (compared to young) for example in our previous study (Baruch et al., Science, 2014). This is now better explained in the text (p.9).

Further, the link between IFN-B dependent MEF2C down-regulation to and changes in synapse numbers and learning/memory behaviors are correlative. More evidence to support this link would strengthen the paper.

We now clarified that we do not propose a direct link between the Mef2C downregulation and loss of synapse numbers and learning/memory behaviors. We specifically point to the role of Mef2C in restraining microglial responses to pro-

inflammatory stimuli, which we showed experimentally in Fig. 6 (p.9-10), and suggest that it may contribute to further dysregulation of microglial function in an aged brain, and exacerbation of pro-inflammatory conditions (p.11-12), which have an overall negative effect on cognitive ability (Guinta, J.Neuroinflammation, 2008).

Moreover does TNF-a injection also alter synapse numbers or learning/memory behavior?

We quantified synaptic terminals (the figure shows Synaptophysin staining intensity in CA3) in TNF-a-injected Mef2C-KO and Mef2C-Ctrl mice, but did not observe any difference.

We did not perform learning and memory tests, routinely used in aged mice, because they require at least two sessions (training and testing) separated by 24h (readout 48h from the moment of treatment), whereas the effect of TNFα treatment in wild type mice resolves within <48h. For example, we observed that levels of TNF-a expression in microglia of Mef2C-KO and Mef2C-Ctrl mice were equal, even though the Mef2C-KO mice showed marked elevation of TNF-a at the 24h time-point (Fig. 6 in the

manuscript).

Minor comments

4. The mouse C4a gene (otherwise known as SLP) is not homologous to human C4a (an isoform of complement component C4 implicated in risk for schizophrenia). The only C4 component that mice have is C4b. Thus, it is confusing how the authors can use an anti-mouse “C4a/b” antibody (Supplemental Figure 9) and claim that they are staining for mouse complement component C4. Has this antibody been tested in mouse C4 KO tissue? Please clarify.

We would like to thank the reviewer for this comment. According to our sequencing alignment pipeline (mouse NCBI/mm10) and gene annotation, the genes expressed in microglia are C4a and C4b, as formally defined according to the MGI and NCBI-databases (the databases mention that the synonym is Slp and C4, but the formal names are C4a and 4b, respectively).

In our immunohistochemistry, we used an antibody directed against the human C4a/b protein (Bioss; bs-11274R). This antibody cross-reacts with the mouse C4, based on the Bioss data (<https://www.biossusa.com/products/bs-11274r#image-tab>). Importantly, according to the literature C4 was implicated in synaptic pruning in mice (Sekar et al, Nat.Med., 2016). We now change the name of the protein to C4 in the text (p.7-8), online methods (p.5), supplementary figures 9 and 13, and the corresponding legends.